# ENTROPIC RISK-SENSITIVE REINFORCEMENT LEARNING: A META REGRET FRAMEWORK WITH FUNCTION APPROXIMATION

## ABSTRACT

We study risk-sensitive reinforcement learning with the entropic risk measure and function approximation. We consider the finite-horizon episodic MDP setting, and propose a meta algorithm based on value iteration. We then derive two algorithms for linear and general function approximation, namely RSVI.L and RSVI.G, respectively, as special instances of the meta algorithm. We illustrate that the success of RSVI.L depends crucially on carefully designed feature mapping and regularization that adapt to risk sensitivity. In addition, both RSVI.L and RSVI.G maintain risk-sensitive optimism that facilitates efficient exploration. On the analytic side, we provide regret analysis for the algorithms by developing a meta analytic framework, at the core of which is a risk-sensitive optimism condition. We show that any instance of the meta algorithm that satisfies the condition yields a meta regret bound. We further verify the condition for RSVI.L and RSVI.G under respective function approximation settings to obtain concrete regret bounds that scale sublinearly in the number of episodes.

## 1 INTRODUCTION

Risk is one of the most important considerations in decision making, so should it be in reinforcement learning (RL). As a prominent paradigm in RL that performs learning while accounting for risk, *risk-sensitive RL* explicitly models risk of decisions via certain risk measures and optimizes for rewards simultaneously. It is poised to play an essential role in application domains where accounting for risk in decision making is crucial. A partial list of such domains includes autonomous driving (Buehler et al., 2009; Thrun, 2010), behavior modeling (Niv et al., 2012; Shen et al., 2014), real-time strategy games (Berner et al., 2019; Vinyals et al., 2019) and robotic surgery (Fagogenis et al., 2019; Shademan et al., 2016).

In this paper, we study risk-sensitive RL through the lens of function approximation, which is an important apparatus for scaling up and accelerating RL algorithms in applications of high dimension. We focus on risk-sensitive RL with the entropic risk measure, a classical framework established by the seminal work of Howard & Matheson (1972). Informally, for a fixed risk parameter $\beta \neq 0$, our goal is to maximize the objective

$$V_\beta = \frac{1}{\beta} \log\{\mathbb{E}e^{\beta R}\}. \tag{1}$$

The definition of $V_\beta$ will be made formal later in (2). The objective (1) admits a Taylor expansion $V_\beta = \mathbb{E}[R] + \frac{\beta}{2}\text{Var}(R) + O(\beta^2)$. Comparing (1) with the risk-neutral objective $V = \mathbb{E}[R]$ studied in the standard RL setting, we see that $\beta > 0$ induces a risk-seeking objective and $\beta < 0$ induces a risk-averse one. Therefore, the formulation with the entropic risk measure in (1) accounts for both risk-seeking and risk-averse modes of decision making, whereas most others are restricted to the risk-averse setting (Fu et al., 2018). It can also be seen that $V_\beta$ tends to the risk-neutral $V$ as $\beta \to 0$.

Existing works on function approximation for RL have mostly focused on the risk-neutral setting and heavily exploits the linearity of risk-neutral objective $V$ in both transition dynamics (implicitly captured by the expectation) and the reward $R$, which is clearly not available in the risk-sensitive objective (1). It is also well known that even in the risk-neutral setting, improperly implemented function approximation could result in errors that scale exponentially in the size of the state space.

Combined with nonlinearity of the risk-sensitive objective (1), it compounds the difficulties of implementing function approximation in risk-sensitive RL with provable guarantees.

This work provides a principled solution to function approximation in risk-sensitive RL by overcoming the above difficulties. Under the finite-horizon MDP setting, we propose a meta algorithm based on value iteration, and from that we derive two concrete algorithms for linear and general function approximation, which we name RSVI.L and RSVI.G, respectively. By modeling a shifted exponential transformation of estimated value functions, RSVI.L and RSVI.G cater to the nonlinearity of the risk-sensitive objective (1) and adapt to both risk-seeking and risk-averse settings. Moreover, both RSVI.L and RSVI.G maintain risk-sensitive optimism in the face of uncertainty for effective exploration. In particular, RSVI.L exploits a synergistic relationship between feature mapping and regularization in a risk-sensitive fashion. The resulting structure of RSVI.L makes it more efficient in runtime and memory than RSVI.G under linear function approximation, while RSVI.G is more general and allows for function approximation beyond the linear setting.

Furthermore, we develop a meta regret analytic framework and identify a risk-sensitive optimism condition that serves as the core component of the framework. Under the optimism condition, we prove a meta regret bound incurred by any instance of the meta algorithm, regardless of function approximation settings. Furthermore, we show that both RSVI.L and RSVI.G satisfy the optimism condition under the respective function approximation and achieve regret that scales sublinearly in the number of episodes. The meta framework therefore helps us disentangle the analysis associated with function approximation from the generic analysis, shedding light on the role of function approximation in regret guarantees. We hope that our meta framework will motivate and benefit future studies of function approximation in risk-sensitive RL.

**Our contributions.** We may summarize the contributions of the present paper as follows:

- we study function approximation in risk-sensitive RL with the entropic risk measure; we provide a meta algorithm, from which we derive two concrete algorithms for linear and general function approximation, respectively; the concrete algorithms are both shown to adapt to all levels of risk sensitivity and maintain risk-sensitive optimism over the learning process;

- we develop a meta regret analytic framework and identify a risk-sensitive optimism condition, under which we prove a meta regret bound for the meta algorithm; furthermore, by showing that the optimism condition holds for both concrete algorithms, we establish regret bounds for them under linear and general function approximation, respectively.

**Notations.** For a positive integer $n$, we let $[n] \coloneqq \{1, 2, \ldots, n\}$. For a number $u \neq 0$, we define $\mathrm{sign}(u) = 1$ if $u > 0$ and $-1$ if $u < 0$. For two non-negative sequences $\{a_i\}$ and $\{b_i\}$, we write $a_i \lesssim b_i$ if there exists a universal constant $C > 0$ such that $a_i \leq C b_i$ for all $i$, and write $a_i \asymp b_i$ if $a_i \lesssim b_i$ and $b_i \lesssim a_i$. We use $\tilde{O}(\cdot)$ to denote $O(\cdot)$ while hiding logarithmic factors. For any $\varepsilon > 0$ and set $\mathcal{X}$, we let $\mathcal{N}_\varepsilon(\mathcal{X}, \|\cdot\|)$ be the $\varepsilon$-net of the set $\mathcal{X}$ with respect to the norm $\|\cdot\|$. We let $\Delta(\mathcal{X})$ be the set of probability distributions supported on $\mathcal{X}$. For any vector $u \in \mathbb{R}^n$ and symmetric and positive definite matrix $\Gamma \in \mathbb{R}^{n \times n}$, we let $\|u\|_\Gamma \coloneqq \sqrt{u^\top \Gamma u}$. We denote by $I_n$ the $n \times n$ identity matrix.

## 2 RELATED WORK

Initiated by the seminal work of Howard & Matheson (1972), risk-sensitive control/RL with the entropic risk measure has been studied in a vast body of literature (Bäuerle & Rieder, 2014; Borkar, 2001; 2002; 2010; Borkar & Meyn, 2002; Cavazos-Cadena & Hernández-Hernández, 2011; Coraluppi & Marcus, 1999; Di Masi & Stettner, 1999; 2000; 2007; Fleming & McEneaney, 1995; Hernández-Hernández & Marcus, 1996; Jaśkiewicz, 2007; Marcus et al., 1997; Mihatsch & Neuneier, 2002; Osogami, 2012; Patek, 2001; Shen et al., 2013; 2014; Whittle, 1990). Yet, this line of works either assumes known transition kernels or focuses on asymptotic behaviors of the problem/algorithms, and finite-sample/time results with unknown transitions have rarely been investigated.

The most relevant work to ours is perhaps Fei et al. (2020), who consider the same problem as ours under the tabular setting. They propose two algorithms based on value iteration and Q-learning. They prove regret bounds for their algorithms, which are then certified to be nearly optimal by a lower bound. However, their algorithms and analysis are restricted to the tabular setting. Compared to Fei et al. (2020), our paper provides a novel and unified framework of algorithms and analysis for function approximation. We study linear and general function approximation as two instances of the framework, both of which subsume the tabular setting.

We also briefly discuss existing works on function approximation with regret analysis, which so far have focused on the risk-neutral setting. The works of Cai et al. (2019); Jin et al. (2019); Wang et al. (2019); Yang & Wang (2019); Zhou et al. (2020) study linear function approximation, while Ayoub et al. (2020); Wang et al. (2020) investigate general function approximation. In addition, all these works prove $\tilde{O}(K^{1/2})$-regret for their algorithms, although dependence on other parameters varies in settings. As we have argued in the previous section, the nonlinear objective (1) makes algorithm design and regret analysis for function approximation much more challenging in risk-sensitive settings than in the standard risk-neutral one.

## 3 PROBLEM FORMULATION

### 3.1 EPISODIC MDP

An episodic MDP is parameterized by a tuple $(K, H, \mathcal{S}, \mathcal{A}, \{P_h\}_{h \in [H]}, \{r_h\}_{h \in [H]})$, where $K$ is the number of episodes, $H$ is the number of steps in each episode, $\mathcal{S}$ is the state space, $\mathcal{A}$ is the action space, $P_h : \mathcal{S} \times \mathcal{A} \to \Delta(\mathcal{S})$ is the transition kernel at step $h$, and $r_h : \mathcal{S} \times \mathcal{A} \to [0, 1]$ is the reward function at step $h$. We assume that $\{P_h\}$ are unknown. For simplicity we also assume that $\{r_h\}$ are known and deterministic, as is done in existing works such as Yang & Wang (2019); Zhou et al. (2020).

We interact with the episodic MDP as follows. In the beginning of each episode $k \in [K]$, the environment chooses an arbitrary initial state $s_1^k \in \mathcal{S}$. Then in each step $h \in [H]$, we take an action $a_h^k \in \mathcal{A}$, receives a reward $r_h(s_h^k, a_h^k)$ and transitions to the next state $s_{h+1}^k \in \mathcal{S}$ sampled from $P_h(\cdot \mid s_h^k, a_h^k)$. Once we reach $s_{H+1}^k$, the current episode terminates and we advance to the next episode unless $k = K$.

### 3.2 VALUE FUNCTIONS, BELLMAN EQUATIONS AND REGRET

We assume that $\beta$ is fixed prior to the learning process, and for notational simplicity we omit it from quantities to be introduced subsequently. In risk-sensitive RL with the entropic risk measure, we aim to find a policy $\pi = \{\pi_h : \mathcal{S} \to \mathcal{A}\}$ so as to maximize the *value function* given by

$$V_h^\pi(s) \coloneqq \frac{1}{\beta} \log \left\{ \mathbb{E} \left[ \exp \left( \beta \sum_{h'=h}^H r_{h'}(s_{h'}, \pi_{h'}(s_{h'})) \right) \right] \Bigg| s_h = s \right\}, \tag{2}$$

for all $(h, s) \in [H] \times \mathcal{S}$. Under some mild regularity conditions, there exists a greedy policy $\pi^* = \{\pi_h^*\}$ which gives the optimal value $V_h^{\pi^*}(s) = \sup_\pi V_h^\pi(s)$ for all $(h, s) \in [H] \times \mathcal{S}$ (Bäuerle & Rieder, 2014). In addition to the value function, another key notion is the *action-value function* defined as

$$Q_h^\pi(s, a) \coloneqq \frac{1}{\beta} \log \left\{ \mathbb{E} \left[ \exp \left( \beta \sum_{h'=h}^H r_{h'}(s_{h'}, a_{h'}) \right) \right] \Bigg| s_h = s, a_h = a \right\}, \tag{3}$$

for all $(h, s, a) \in [H] \times \mathcal{S} \times \mathcal{A}$.

The action-value function $Q_h^\pi$ is closely associated with the value function $V_h^\pi$ via the so-called *Bellman equation*:

$$Q_h^\pi(s, a) = r_h(s, a) + \frac{1}{\beta} \log \left\{ \mathbb{E}_{s' \sim P_h(\cdot \mid s, a)} \left[ \exp \left( \beta \cdot V_{h+1}^\pi(s') \right) \right] \right\}, \tag{4}$$

$$V_h^\pi(s) = Q_h^\pi(s, \pi_h(s)), \qquad V_{H+1}^\pi(s) = 0,$$

which holds for all $(h, s, a) \in [H] \times \mathcal{S} \times \mathcal{A}$. Note that the identity of $Q_h^\pi$ in (4) is a result of simple calculation based on (2) and (3). Similarly, the *Bellman optimality equation* is given by

$$Q_h^*(s, a) = r_h(s, a) + \frac{1}{\beta} \log \left\{ \mathbb{E}_{s' \sim P_h(\cdot \mid s, a)} \left[ \exp \left( \beta \cdot V_{h+1}^*(s') \right) \right] \right\}, \tag{5}$$
$$V_h^*(s) = \max_{a \in \mathcal{A}} Q_h^*(s, a), \qquad V_{H+1}^*(s) = 0,$$

again for all $(h, s, a) \in [H] \times \mathcal{S} \times \mathcal{A}$. In the above, we use the shorthand $Q_h^*(\cdot, \cdot) := Q_h^{\pi^*}(\cdot, \cdot)$ for all $h \in [H]$ and $V_h^*(\cdot)$ is similarly defined. The identity $V_h^*(\cdot) = \max_{a \in \mathcal{A}} Q_h^*(\cdot, a)$ implies that the optimal $\pi^*$ is the greedy policy with respect to the optimal action-value function $\{Q_h^*\}_{h \in [H]}$.

During the learning process, the policy $\pi^k$ in each episode $k$ may be different from the optimal $\pi^*$. We quantify this difference over all $K$ episodes through the notion of *regret*, defined as

$$\text{Regret}(K) := \sum_{k \in [K]} \left[ V_1^*(s_1^k) - V_1^{\pi^k}(s_1^k) \right]. \tag{6}$$

Since $V_1^*(s) \geq V_1^\pi(s)$ for any $\pi$ and $s \in \mathcal{S}$ by definition, regret also characterizes the suboptimality of $\{\pi^k\}$ relative to the optimal $\pi^*$.

### 3.3 FUNCTION APPROXIMATION

In this paper, we focus on linear and general function approximation. We consider the following form of linear function approximation, which assumes that each transition kernel admits a linear form.

**Assumption 1** *We assume that the MDP is equipped with a known feature function $\psi : \mathcal{S} \times \mathcal{A} \times \mathcal{S} \to \mathbb{R}^d$ such that for any $h \in [H]$, there exists a vector $\theta_h \in \mathbb{R}^d$ with $\|\theta_h\|_2 \leq \sqrt{d}$ and the transition kernel is given by*

$$P_h(s' \mid s, a) = \psi(s, a, s')^\top \theta_h$$

*for any $(s, a, s') \in \mathcal{S} \times \mathcal{A} \times \mathcal{S}$. We also assume that*

$$\left\| \int_{\mathcal{S}} \psi(s, a, s') \cdot (e^{\beta \cdot V(s')} - 1) \mathrm{d}s' \right\|_2 \leq \sqrt{d} \cdot \left| e^{\beta \bar{v}} - 1 \right|,$$

*for any $(s, a) \in \mathcal{S} \times \mathcal{A}$ and $V : \mathcal{S} \to [0, \bar{v}]$ where $\bar{v} \geq 0$.*

This form of linear function approximation is also studied in the work of Ayoub et al. (2020); Cai et al. (2019); Zhou et al. (2020), whose setting is equivalent to ours when $\beta \to 0$. The setting of Assumption 1 may be reduced to the tabular setting in which $\psi(s, a, s')$ is a canonical basis vector in $\mathbb{R}^d$ with $d = |\mathcal{S}|^2 |\mathcal{A}|$, i.e., the $(s, a, s')$-th entry of $\psi(s, a, s')$ is equal to one and the other entries are equal to zero. It also subsumes various settings of function approximation including linear combinations of base models (Modi et al., 2020) and the matrix bandit setting (Yang & Wang, 2019); we refer readers to (Zhou et al., 2020) for more details on the generality of Assumption 1.

For general function approximation, we make the following assumption.

**Assumption 2** *We assume that we have access to a function set[1] $\mathcal{P}$ such that the transition kernel $P_h \in \mathcal{P}$ for all $h \in [H]$.*

This setting is also considered in Ayoub et al. (2020). Under Assumption 2, we may measure the complexity of function sets using the notion of the so-called eluder dimension (Russo & Van Roy, 2014), which we define and discuss in Appendix A.

Note that although we focus on function approximation of transition kernels, a similar approach can be taken to apply function approximation to reward functions and our regret guarantees presented below would still hold, as argued in Yang & Wang (2019).

---

[1] Throughout the paper, we use *function class* and *function set* interchangeably.

## 4 ALGORITHMS

We first present the Meta algorithm for Risk-Sensitive Value Iteration (MetaRSVI) in Algorithm 1, which is a high-level framework including key features of algorithms based on value iteration (Bradtke & Barto, 1996; Osband et al., 2014; Jin et al., 2019). It mainly consists of a value estimation step and policy execution step. In the value estimation step, the algorithm estimates the optimal $\{Q_h^*\}$ by its iterates $\{Q_h^k\}$ based on historical data. Since we focus on greedy policies, in Line 5 the estimated value function $V_h^k(\cdot)$ is simply taken as the maximum among $\{Q_h^k(\cdot, a')\}_{a' \in \mathcal{A}}$. The primary machinery of value estimation, known as Risk-Sensitive Temporal Difference or RSTD, is presented in an abstract way in Line 4; this is because its concrete form would depend on function approximation of the underlying MDP and algorithmic implementation. In the policy execution step, the algorithm uses the policy learned so far (represented by $\{Q_h^k\}$) to collect data for subsequent learning stages. We remark that Algorithm 1 is flexible and general enough to allow for any function approximation. In the remaining of this section, we derive two special instances of Algorithm 1 for linear and general function approximation, respectively, by providing concrete implementation of RSTD.

---

**Algorithm 1** MetaRSVI

**Input:** risk parameter $\beta$, number of episodes $K$
1: **for** episode $k = 1, \ldots, K$ **do**
2:      $V_{H+1}^k(\cdot) \leftarrow 0$
3:      **for** step $h = H, H-1, \ldots, 1$ **do**                  ▷ *value estimation*
4:          $Q_h^k(\cdot, \cdot) \leftarrow \text{RSTD}(k, h, \beta, \{V_{h+1}^\tau\}_{\tau \in [k]})$
5:          $V_h^k(\cdot) \leftarrow \max_{a' \in \mathcal{A}} Q_h^k(\cdot, a')$
6:      **end for**
7:      Receive the initial state $s_1^k$ from the environment
8:      **for** step $h = 1, 2, \ldots, H$ **do**                        ▷ *policy execution*
9:          Take action $a_h^k \leftarrow \text{argmax}_{a' \in \mathcal{A}} Q_h^k(s_h^k, a')$
10:        Receive the reward $r_h(s_h^k, a_h^k)$ and the next state $s_{h+1}^k$
11:      **end for**
12: **end for**

---

We introduce Risk-Sensitive Value Iteration for Linear function approximation, or RSVI.L, in Algorithm 2 under the setting of Assumption 1. This algorithm is inspired by RSVI proposed in Fei et al. (2020) that specializes in the tabular setting. It replaces the abstract function RSTD in Algorithm 1 with a concrete implementation for the linear function approximation. In Line 4 the iterate $w_h^k$ can be interpreted as the solution of the following least-squares problem:

$$w_h^k \leftarrow \underset{w \in \mathbb{R}^d}{\text{argmin}} \sum_{\tau \in [k-1]} [e^{\beta \cdot V_{h+1}^\tau(s_{h+1}^\tau)} - 1 - w^\top \phi_h^\tau(s_h^\tau, a_h^\tau)]^2 + \lambda \|w\|_2^2, \tag{7}$$

where the surrogate feature mappings $\{\phi_h^\tau(\cdot, \cdot)\}_{\tau \in [k-1]}$ are constructed in Line 5 and $\lambda \geq 0$ is the regularization parameter to be set by users. The above regression problem essentially computes an estimate of $\theta_h$, the parameter of the transition kernel $P_h$, by taking advantage of the linear form $\mathbb{E}_{s' \sim P_h(\cdot \mid s, a)}[e^{\beta \cdot V_{h+1}^\pi(s')}]$ in the Bellman equation (4). Note that the regression targets are set as a shifted exponential transformation of estimated value functions, i.e., $\{e^{\beta \cdot V_{h+1}^\tau(s_{h+1}^\tau)} - 1\}$. Similar construction is applied to the surrogate features $\phi_h^k(\cdot, \cdot)$ in Line 5. Mechanically, such design ensures that when $V_{h+1}^k(\cdot) = 0$, we would have $\phi_h^k(\cdot, \cdot) = 0$ and therefore by definition $Q_h^k(\cdot, \cdot) = r_h(\cdot, \cdot)$; this is a similar behavior exhibited by risk-neutral algorithms for linear function approximation, e.g., Algorithm 1 in Cai et al. (2019) whose surrogate features are given by replacing $e^{\beta \cdot V_{h+1}^k(\cdot)} - 1$ in Line 5 with $V_{h+1}^k(\cdot)$. To update $Q_h^k(\cdot, \cdot)$ in Line 7, we use the quantity

$$q_{h,L}^k(\cdot, \cdot) := \begin{cases} \min\{e^{\beta(H-h)}, \langle \phi_h^k(\cdot, \cdot), w_h^k \rangle + 1 + b_h^k(\cdot, \cdot)\}, & \text{if } \beta > 0, \\ \max\{e^{\beta(H-h)}, \langle \phi_h^k(\cdot, \cdot), w_h^k \rangle + 1 - b_h^k(\cdot, \cdot)\}, & \text{if } \beta < 0. \end{cases} \tag{8}$$

Informally, with $b_h^k$ taking the role of bonus, $q_{h,L}^k$ can be seen as an "optimistic" and risk-adaptive estimate for the expected value of $e^{\beta \cdot V_{h+1}^k(\cdot)}$ under the (unknown) transition kernel $P_h$. The term

---

**Algorithm 2** RSVI.L

---

**Input:** risk parameter $\beta$, number of episodes $K$, bonus multiplier $\gamma_L$, regularization parameter $\lambda$

1: Run Algorithm 1 with RSTD() therein overloaded by the following subroutine:
2: **procedure** RSTD($k, h, \beta, \{V_{h+1}^\tau\}_{\tau \in [k]}, \gamma_L, \lambda$)
3:     $\Lambda_h^k \leftarrow \sum_{\tau \in [k-1]} \phi_h^\tau(s_h^\tau, a_h^\tau) \phi_h^\tau(s_h^\tau, a_h^\tau)^\top + \lambda I_d$
4:     $w_h^k \leftarrow (\Lambda_h^k)^{-1} \sum_{\tau \in [k-1]} \phi_h^\tau(s_h^\tau, a_h^\tau) \cdot (e^{\beta \cdot V_{h+1}^\tau(s_{h+1}^\tau)} - 1)$
5:     $\phi_h^k(\cdot, \cdot) \leftarrow \int_\mathcal{S} \psi(\cdot, \cdot, s') \cdot (e^{\beta \cdot V_{h+1}^k(s')} - 1) \mathrm{d}s'$
6:     $b_h^k(\cdot, \cdot) \leftarrow \gamma_L \cdot [\phi_h^k(\cdot, \cdot)^\top (\Lambda_h^k)^{-1} \phi_h^k(\cdot, \cdot)]^{1/2}$
7:     **return** $Q_h^k(\cdot, \cdot) \leftarrow r_h(\cdot, \cdot) + \frac{1}{\beta} \log(q_{h,L}^k(\cdot, \cdot))$, where $q_{h,L}^k(\cdot, \cdot)$ is defined in (8)
8: **end procedure**

---

$\langle \phi_h^k(\cdot, \cdot), w_h^k \rangle + 1$ serves as a correction of the shifted quantity $e^{\beta \cdot V_{h+1}^k(\cdot)} - 1$ in both $\phi_h^k(\cdot, \cdot)$ and $w_h^k$, so as to align the structure of Line 7 to that of the Bellman equation (4). A proper definition of bonus $b_h^k$ (with $\gamma_L$ therein formally given in Theorem 2 below) and the truncation at $e^{\beta(H-h)}$ would put $q_{h,L}^k$ within $[1, e^{\beta(H-h)}]$ entrywise. This ensures that the estimate $Q_h^k(\cdot, \cdot)$ is on the same scale as the optimal $Q_h^*(\cdot, \cdot) \in [0, H-h+1]$.

We note that there is a significant difference between Algorithm 2 and RSVI introduced in Fei et al. (2020). Since RSVI is designed only for the tabular setting, it suffices to set $\lambda = 0$ and update the entries in $Q_h^k$ whose corresponding state-action pairs have been visited in the history. In the linear setting, however, the entire $Q_h^k$ is updated at once and it's imperative to prevent the singularity of the covariance matrix $\Lambda_h^k$. This boils down to having a proper choice of the regularization parameter $\lambda$. If $\lambda$ is too small, $\Lambda_h^k$ would be nearly singular; if $\lambda$ is too large, the spectrum of $\Lambda_h^k$ would be dominated by $\lambda$ for prohibitively many episodes with the algorithm making little progress in learning. Intuitively, since $\|\phi_h^\tau(\cdot, \cdot)\|_2 \propto |e^{\beta H} - 1|$, an ideal $\lambda$ should depend on $\beta$ and be on the order of $(e^{\beta H} - 1)^2$, so that the spectrums of the matrices $\phi_h^\tau(\cdot, \cdot) \phi_h^\tau(\cdot, \cdot)^\top$ and $\lambda I_d$ are close. Indeed, this intuition provides guidance to our choice of $\lambda$ (see Theorem 2). Our Algorithm 2 hence demonstrates a synergistic relationship between the surrogate features and regularization in a risk-sensitive fashion. This is in great contrast with existing algorithms for linear function approximation in the risk-neutral setting, such as Cai et al. (2019); Zhou et al. (2020), in which the design of surrogate features and choice of regularization are decoupled.

For general function approximation under Assumption 2, we present Risk-Sensitive Value Iteration for General function approximation, abbreviated as RSVI.G, in Algorithm 3 of Appendix B. Despite their apparent difference, Algorithms 2 and 3 both implement the principle of Risk-Sensitive Optimism in the Face of Uncertainty (RS-OFU) (Fei et al., 2020), by adding bonus/maximizing over the confidence set when $\beta > 0$, and subtracting bonus/minimizing over the confidence set when $\beta < 0$. Such mechanism encourages the algorithms to explore actions that may have rarely been taken due to low estimated Q-values, while accounting for risk sensitivity at the same time.[2]

## 5 MAIN RESULTS

In this section, we present regret guarantees for our algorithms via a meta regret framework. We identify a risk-sensitive optimism condition, which certifies a certain form of optimism of an algorithm. Under the condition, we first provide a meta regret bound for Algorithm 1. We then instantiate the meta regret bound for Algorithms 2 and 3 under Assumptions 1 and 2, respectively.

### 5.1 META REGRET BOUND

Recall the iterates $\{Q_h^k\}$ in Algorithm 1. For each tuple $(k, h, s, a) \in [K] \times [H] \times \mathcal{S} \times \mathcal{A}$, we define $\overline{Q}_h^k(s, a) := r_h(s, a) + \frac{1}{\beta} \log\{\mathbb{E}_{s' \sim P_h(\cdot \,|\, s,a)} e^{\beta \cdot V_{h+1}^k(s')}\}$. It can be seen that $\{\overline{Q}_h^k\}$ are the

---

[2]We also discuss computational aspects of our algorithms; see Appendix B.

ideal counterparts of $\{Q_h^k\}$ that could be constructed if the transition kernels $\{P_h\}$ were known. We set forth the following condition, which is the central component of our meta regret framework.

**Condition 1** *For all $(k, h, s, a) \in [K] \times [H] \times \mathcal{S} \times \mathcal{A}$, we have $Q_h^k(s, a) \in [0, H - h + 1]$, and there exist some quantities $m_h^k(s, a) > 0, g \geq 1$ and universal constant $c > 0$ such that*

$$0 \leq Q_h^k(s, a) - \overline{Q}_h^k(s, a) \leq c \cdot \frac{e^{|\beta|H} - 1}{|\beta|} \cdot g \cdot m_h^k(s, a).$$

Since $\{Q_h^k\}$ are informally "optimistic" estimates of the ideal $\{\overline{Q}_h^k\}$, the difference $Q_h^k(s, a) - \overline{Q}_h^k(s, a)$ in Condition 1 may be thought of as the level of optimism maintained by the algorithm for state-action pair $(s, a)$ in step $h$ of episode $k$, with its upper bound depending on risk sensitivity through the factor $\frac{e^{|\beta|H} - 1}{|\beta|}$. Therefore, we say that Condition 1 is a risk-sensitive optimism condition. In the upper bound of the condition, the actual values of $g$ and $m_h^k(s, a)$ may depend on function approximation and implementation of the abstract function RSTD. Let us recall that $\{(s_h^k, a_h^k)\}$ are the state-action pairs visited by Algorithm 1, and we are ready to state the meta regret bound.

**Theorem 1** *Let $M := g \sum_{k \in [K]} \sum_{h \in [H]} \min\{1, m_h^k(s_h^k, a_h^k)\}$, where $g$ and $\{m_h^k\}$ are as given in Condition 1. On the event of Condition 1, for any $\delta \in (0, 1]$, with probability at least $1 - \delta$ the regret of Algorithm 1 satisfies*

$$Regret(K) \lesssim \frac{e^{|\beta|H} - 1}{|\beta|} e^{|\beta|H^2} M + e^{|\beta|H^2} \sqrt{KH^3 \log(1/\delta)}.$$

The proof is given in Appendix D. Even though the actual form of $M$ depends on specific function approximation, the derivation of Theorem 1 only requires the structure of Algorithm 1, which is agnostic of function approximation. In the above bound, the first term can be interpreted as the total optimism maintained by Algorithm 1, and is in fact a direct consequence of Condition 1. The second term can be seen as the total drift of iterates $\{V_h^k\}$ from the value functions $\{V_h^{\pi_k}\}$, which is the result of a martingale analysis. The factor $e^{|\beta|H^2}$ shared by both terms is due to a local linearization of the nonlinear objective (2) as well as a standard backward induction analysis of $H$-horizon MDPs. Soon we will show that $M = \tilde{O}(K^{1/2})$ under both linear and general function approximation, so the first term in Theorem 1 would dominate in the regret bound. Similar to $M$, the exponential factor $\frac{e^{|\beta|H} - 1}{|\beta|}$ also comes into the bound from Condition 1. It has been shown as a distinctive feature of risk-sensitive RL algorithms that represents a tradeoff between risk sensitivity and sample complexity; see Fei et al. (2020) for a detailed discussion on this point.

## 5.2 Regret bound for linear function approximation

We now present a regret bound for Algorithm 2 induced by Theorem 1. Let

$$\gamma_L = c_\gamma |e^{\beta H} - 1| \sqrt{d \log(2dKH/\delta)}, \tag{9}$$

where $c_\gamma > 0$ is an appropriate universal constant. We have the following result.

**Theorem 2** *Let $\gamma_L$ of (9) and $\lambda = (e^{\beta H} - 1)^2$ be input to Algorithm 2, and $M$ be as defined in Theorem 1. Under Assumption 1, for any $\delta \in (0, 1]$, with probability at least $1 - \delta$, Condition 1 holds for Algorithm 2 so that $M \lesssim [d^2 K H^2 \log^2(2dKH/\delta)]^{1/2}$. Therefore, Theorem 1 implies that the regret of Algorithm 2 satisfies*

$$Regret(K) \lesssim \frac{e^{|\beta|H} - 1}{|\beta|} e^{|\beta|H^2} \sqrt{d^2 K H^2 \log^2(2dKH/\delta)}.$$

The proof is given in Appendix E. One may obtain a regret bound for the tabular setting by taking $d = |\mathcal{S}|^2 |\mathcal{A}|$ in Theorem 2, and the resulting bound can be seen to nearly match that of Fei et al. (2020, Theorem 1), except for the polynomial dependency on $|\mathcal{S}|$ and $|\mathcal{A}|$.[3] The bound obtained

---

[3]By inspecting the proof of Fei et al. (2020, Theorem 1), we see that they apply the bound $(1/|\beta|)(\exp(|\beta| H) - 1) \exp(|\beta| H^2) \leq (1/|\beta|)(\exp(C |\beta| H^2) - 1)$ for some universal constant $C > 0$.

from specializing Theorem 2 to the tabular setting is also nearly optimal for small $|\beta|$ (with respect to $|\beta|$, $K$ and $H$) in view of the lower bound

$$\mathbb{E}\left[\text{Regret}(K)\right] \gtrsim \frac{e^{|\beta|H/2} - 1}{|\beta|} \sqrt{K \log K} \tag{10}$$

given by Fei et al. (2020, Theorem 3). In addition, as $\beta \to 0$, the setting of risk-sensitive RL tends to that of standard risk-neutral RL. We have the following corollary as a precise characterization of Theorem 2 in that regime.

**Corollary 1** *Under the setting of Theorem 2 and when $\beta \to 0$, with probability at least $1 - \delta$, the regret of Algorithm 2 satisfies*

$$Regret(K) \lesssim \sqrt{d^2 K H^4 \log^2(2dKH/\delta)}.$$

The proof is given in Appendix F. Corollary 1 matches the standard result in the risk-neutral setting, e.g. Cai et al. (2019, Theorem 3.1), up to logarithmic factors.

## 5.3 REGRET BOUND FOR GENERAL FUNCTION APPROXIMATION

To present the regret guarantee for general function approximation, we need to set a few additional notations. Recall the function set $\mathcal{P}$ from Assumption 2. For any $P \in \mathcal{P}$, $(s, a) \in \mathcal{S} \times \mathcal{A}$ and $V : \mathcal{S} \to [0, H]$, we define the function set

$$\mathcal{Z} := \{z_P : P \in \mathcal{P}\}, \quad \text{where } z_P(s, a, V) := \text{sign}(\beta) \int_{\mathcal{S}} P(s' \mid s, a) \cdot (e^{\beta \cdot V(s')} - 1) \mathrm{d}s', \tag{11}$$

For any $P, P' \in \mathcal{P}$, we define $\|P - P'\|_{\infty, 1} := \sup_{(s,a) \in \mathcal{S} \times \mathcal{A}} \|P(\cdot \mid s, a) - P'(\cdot \mid s, a)\|_1$. We let

$$d_E := \dim_E(\mathcal{Z}, |e^{\beta H} - 1|/K)$$

be the $(|e^{\beta H} - 1|/K)$-eluder dimension of function set $\mathcal{Z}$,[4] and

$$\zeta := \log\left(H \cdot \mathcal{N}_{1/K}(\mathcal{P}, \|\cdot\|_{\infty, 1})/\delta\right) + \sqrt{\log(4K^2 H/\delta)}. \tag{12}$$

In Algorithm 3, we set

$$\gamma_G = 10|e^{\beta H} - 1|\sqrt{\zeta}. \tag{13}$$

We now state our result for Algorithm 3, which is another instantiation of Theorem 1.

**Theorem 3** *Let $\gamma_G$ of (13) be input to Algorithm 3 and $M$ be as defined in Theorem 1. Under Assumption 2, for any $\delta \in (0, 1]$, with probability at least $1 - \delta$, Condition 1 holds for Algorithm 3 so that $M \lesssim H \min\{d_E, K\} + \sqrt{d_E K H^2 \zeta}$. Therefore, Theorem 1 implies that the regret of Algorithm 3 satisfies*

$$Regret(K) \lesssim \frac{e^{|\beta|H} - 1}{|\beta|} e^{|\beta|H^2} \left(H \min\{d_E, K\} + \sqrt{d_E K H^2 \zeta}\right).$$

The proof is given in Appendix G. The term $H \min\{d_E, K\}$ above also appears in the regret bound for the multi-arm bandit problem with general function approximation (Russo & Van Roy, 2014), which is a special case of our finite-horizon episodic MDP setting. When $K$ is sufficiently large, we have $H \min\{d_E, K\} \lesssim \sqrt{d_E K H^2 \zeta}$ and therefore Theorem 3 yields $\text{Regret}(K) = \frac{e^{|\beta|H} - 1}{|\beta|} e^{|\beta|H^2} \tilde{O}(\sqrt{d_E K H^2})$. In case that the transition kernels in $\mathcal{P}$ take the linear form as in Assumption 1, we have $d_E \lesssim d \log K$, and $\log(\mathcal{N}_{1/K}(\mathcal{P}, \|\cdot\|_{\infty, 1})) \lesssim d \log K$ so that $\zeta \lesssim d \log(KH/\delta)$. Then for sufficiently large $K$, the bound in Theorem 3 matches that in Theorem 2 up to a logarithmic factor. On the other hand, under the linear setting of Assumption 1, Algorithm 2 may be more efficient in runtime and memory than Algorithm 3, as discussed in Appendix B.

---

[4]Recall that the definition of the eluder dimension is formally given in Appendix A.

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

## A   ELUDER DIMENSION

To introduce the eluder dimension, we need to set forth the concept of $\varepsilon$-independence.

**Definition 1** *For any $\varepsilon > 0$ and function set $\mathcal{G}$ whose elements are in the domain $\mathcal{X}$, we say that an $x \in \mathcal{X}$ is $\varepsilon$-dependent on the set of elements $\mathcal{X}_n := \{x_1, x_2, \ldots, x_n\} \subset \mathcal{X}$ with respect to $\mathcal{G}$ if any pair of functions $g, g' \in \mathcal{G}$ satisfying $\sum_{i \in [n]} (g(x_i) - g'(x_i))^2 \leq \varepsilon^2$ also satisfies $g(x) - g'(x) \leq \varepsilon$. We say that $x$ is $\varepsilon$-independent of $\mathcal{X}_n$ with respect to $\mathcal{G}$ if $x$ is not $\varepsilon$-dependent on $\mathcal{X}_n$ with respect to $\mathcal{G}$.*

Hence, $\varepsilon$-independence characterizes a notion of dissimilarity of a point $x$ to the elements in subset $\mathcal{X}_n$ of function set $\mathcal{G}$. Now we are ready to formally define the eluder dimension, which quantifies the length of the longest possible chain of dissimilar elements in a function set.

**Definition 2** *For any $\varepsilon > 0$ and function set $\mathcal{G}$ whose elements are in the domain $\mathcal{X}$, the $\varepsilon$-eluder dimension $\dim_E(\mathcal{G}, \varepsilon)$ is defined as the length $d'$ of the longest sequence of elements in $\mathcal{X}$ such that, for some $\varepsilon' \geq \varepsilon$, every element is $\varepsilon'$-independent of its predecessors.*

The eluder dimension extends the concept of dimension in linear spaces and generalizes to non-linear function spaces. It is also related to the notions of Kolmogorov dimension and VC dimension. We refer readers to Russo & Van Roy (2014) for further details on the eluder dimension and its advantages compared to other complexity measures.

## B   MORE ON ALGORITHMS

We present details of Algorithm 3 that we have omitted from the main text. For each $(k, h) \in [K] \times [H]$, transition kernels $P, P' \in \mathcal{P}$ and estimated value functions $\{V_{h+1}^\tau\}_{\tau \in [k-1]}$, we define the squared error

$$\Gamma_h^k(P, P') := \sum_{\tau=1}^{k-1} \left( \int_\mathcal{S} P(s' \mid s_h^\tau, a_h^\tau) e^{\beta \cdot V_{h+1}^\tau(s')} \mathrm{d}s' - \int_\mathcal{S} P'(s' \mid s_h^\tau, a_h^\tau) e^{\beta \cdot V_{h+1}^\tau(s')} \mathrm{d}s' \right)^2. \quad (14)$$

In Algorithm 3, Line 3 computes an estimate $P_h^k$ of the transition kernel $P_h$, by solving a least-squares problem similar to (7). Line 4 then constructs a confidence set around the estimate $P_h^k$. This step is reminiscent of UCRL (Jaksch et al., 2010), where confidence sets are used to enforce the so-called Optimism in the Face of Uncertainty (OFU) principle for efficient exploration. Line 5 updates $Q_h^k$ by solving an optimization problem over the confidence set $\mathcal{P}_h^k$ with the definition

$$q_{h,G}^k(\cdot, \cdot) := \begin{cases} \max_{P \in \mathcal{P}_h^k} \int_\mathcal{S} P(s' \mid \cdot, \cdot) e^{\beta \cdot V_{h+1}^k(s')} \mathrm{d}s', & \text{if } \beta > 0, \\ \min_{P \in \mathcal{P}_h^k} \int_\mathcal{S} P(s' \mid \cdot, \cdot) e^{\beta \cdot V_{h+1}^k(s')} \mathrm{d}s', & \text{if } \beta < 0. \end{cases} \quad (15)$$

It is worth remarking that both Lines 3 and 4 implicitly operate with the shifted exponential transformation of estimated value functions: replacing $e^{\beta \cdot V_{h+1}^\tau(\cdot)}$ therein by $e^{\beta \cdot V_{h+1}^\tau(\cdot)} - 1$ would not result in any difference for the algorithm.

**Computational aspects.**   We briefly discuss computational aspects of Algorithms 2 and 3. It is not hard to see that the time and space complexities for Algorithm 2 are polynomial in key model parameters $d$, $K$ and $H$. For Algorithm 3, it is unclear how the complexities scale under Assumption 2, where the structure of $\mathcal{P}$ is unknown. Nevertheless, under Assumption 1 in which the transition

---

**Algorithm 3** RSVI.G

---

**Input:** risk parameter $\beta$, number of episodes $K$, confidence width $\gamma_G$, function set $\mathcal{P}$
1: Run Algorithm 1 with RSTD() therein overloaded by the following subroutine:
2: **procedure** RSTD($k$, $h$, $\beta$, $\{V_{h+1}^\tau\}_{\tau \in [k]}$, $\gamma_G$, $\mathcal{P}$)
3:     $P_h^k \leftarrow \arg\min_{P \in \mathcal{P}} \sum_{\tau=1}^{k-1} (e^{\beta \cdot V_{h+1}^\tau(s_{h+1}^\tau)} - \int_{\mathcal{S}} P(s' \mid s_h^\tau, a_h^\tau) e^{\beta \cdot V_{h+1}^\tau(s')} ds')^2$
4:     $\mathcal{P}_h^k \leftarrow \{P \in \mathcal{P} : \Gamma_h^k(P, P_h^k) \leq \gamma_G^2\}$, where $\Gamma_h^k(\cdot, \cdot)$ is defined in (14)
5:     **return** $Q_h^k(\cdot, \cdot) \leftarrow r_h(\cdot, \cdot) + \frac{1}{\beta} \log(q_{h,G}^k(\cdot, \cdot))$, where $q_{h,G}^k(\cdot, \cdot)$ is defined in (15)
6: **end procedure**

---

kernels admit a linear form, Algorithm 3 also attains polynomial time and space complexities with respect to key model parameters. The actual runtime and memory consumption of Algorithm 3 may be higher than those of Algorithm 2 though, since the construction of confidence sets in Algorithm 3 requires solving linear programs which could be computationally cumbersome.

## C  PRELIMINARIES TO PROOFS

We fix a tuple $(k, h, s, a) \in [K] \times [H] \times \mathcal{S} \times \mathcal{A}$ and a policy $\pi$. For Algorithm 1 (which subsumes both Algorithms 2 and 3), define

$$q_2 = q_{h,2}^k(s, a) := \mathbb{E}_{s' \sim P_h(\cdot \mid s, a)} e^{\beta \cdot V_{h+1}^k(s')}, \tag{16}$$

and

$$q_3 = q_{h,3}^k(s, a) := \mathbb{E}_{s' \sim P_h(\cdot \mid s, a)} e^{\beta \cdot V_{h+1}^\pi(s')}. \tag{17}$$

In the above definitions, note that $q_2$ and $q_3$ depend on $(k, h, s, a)$; we suppress such dependency for notational simplicity. We have the following bounds on $q_2$ and $q_3$.

**Lemma 1** *We have $q_2, q_3 \in [\min\{1, e^{\beta(H-h)}\}, \max\{1, e^{\beta(H-h)}\}]$.*

*Proof.* The result for $q_2$ and $q_3$ can be seen if we recall their definitions and the fact that $e^{\beta \cdot V(\cdot)} \in [\min\{1, e^{\beta(H-h)}\}, \max\{1, e^{\beta(H-h)}\}]$ for any $V : \mathcal{S} \to [0, H-h]$. $\square$

For any $q' > 0$, define

$$
\begin{aligned}
G_1(q') &:= \frac{1}{\beta} \log\{q'\} - \frac{1}{\beta} \log\{q_2\}, \\
G_2 &:= \frac{1}{\beta} \log\{q_2\} - \frac{1}{\beta} \log\{q_3\}.
\end{aligned}
\tag{18}
$$

Since $q_2, q_3 > 0$ by definition, $G_1(q')$ and $G_2$ are well-defined. By the Bellman equation (4), for any $\pi$ we have

$$Q_h^\pi(s, a) = r_h(s, a) + \frac{1}{\beta} \log\left\{ \mathbb{E}_{s' \sim P_h(\cdot \mid s, a)} e^{\beta \cdot V_{h+1}^\pi(s')} \right\}.$$

We restate Condition 1 in the following.

**Condition 2 (Restatement of Condition 1)** *Let $G_1(q')$ be as defined in (18). For all $(k, h, s, a) \in [K] \times [H] \times \mathcal{S} \times \mathcal{A}$, assume $Q_h^k(s, a) \in [0, H]$ in Algorithm 1, then there exist some quantities $g \geq 1$, $m_h^k = m_h^k(s, a) > 0$, $q_1 = q_{h,1}^k(s, a) \geq \min\{1, e^{\beta(H-h)}\}$ and universal constant $c_1 > 0$ such that $G_1(q_{h,1}^k(s, a)) = Q_h^k(s, a) - \frac{1}{\beta} \log\{q_{h,2}^k(s, a)\}$, and*

$$0 \leq G_1(q_{h,1}^k(s, a)) \leq c_1 \cdot \frac{e^{|\beta|H} - 1}{|\beta|} \cdot g \cdot m_h^k.$$

It can be seen that $G_1(q_{h,1}^k(s, a)) = Q_h^k(s, a) - \overline{Q}_h^k(s, a)$, where $\overline{Q}_h^k$ is defined in Section 5. Under Condition 2, it holds that

$$(Q_h^k - Q_h^\pi)(s, a) = \frac{1}{\beta} \log\{q_1\} - \frac{1}{\beta} \log\{q_3\} = G_1 + G_2, \tag{19}$$

by the construction of $Q_h^k$ in the algorithms, where we have let $G_1 := G_1(q_1)$. Condition 2 has unspecified quantities $q_1$, $\{m_h^k\}$ and $g$. The condition, along with those quantities therein, will be verified in Lemmas 6 and 11 under Assumptions 1 and 2, respectively.

For now let us focus on $G_2$, and we need the following simple result to control it.

**Fact 1** *Consider $x, y, b \in \mathbb{R}$ such that $x \geq y$.*

    *(a) if $y \geq g_0$ for some $g_0 > 0$, then $\log(x) - \log(y) \leq \frac{1}{g}(x - y)$;*

    *(b) Assume further that $y \geq 0$. If $b \geq 0$ and $x \leq u$ for some $u > 0$, then $e^{bx} - e^{by} \leq be^{bu}(x - y)$; if $b < 0$, then $e^{by} - e^{bx} \leq (-b)(x - y)$.*

*Proof.* The results follow from Lipschitz continuity of the functions $x \mapsto \log(x)$ and $x \mapsto e^{bx}$. $\square$

We next control $G_2$, whose proof is agnostic of function approximation.

**Lemma 2** *For each $(k, h, s, a) \in [K] \times [H] \times \mathcal{S} \times \mathcal{A}$, if $V_{h+1}^k(s') \geq V_{h+1}^\pi(s')$ for all $s' \in \mathcal{S}$, then we have*
$$0 \leq G_2 \leq e^{|\beta|H} \cdot \mathbb{E}_{s' \sim P_h(\cdot \mid s, a)}[V_{h+1}^k(s') - V_{h+1}^\pi(s')].$$

*Proof.* **Case $\beta > 0$.** The assumption $V_{h+1}^k(s') \geq V_{h+1}^\pi(s')$ for all $s' \in \mathcal{S}$ implies that $q_2 \geq q_3$ (by the definitions of $q_2$ and $q_3$ in (16) and (17)) and therefore $G_2 \geq 0$ by the definition (18). We also have
$$G_2 \leq \frac{1}{\beta}(q_2 - q_3)$$
$$\leq e^{|\beta|H}\mathbb{E}_{s' \sim P_h(\cdot \mid s, a)}[V_{h+1}^k(s') - V_{h+1}^\pi(s')],$$
where the first step holds by Fact 1(a) (with $g_0 = 1$, $x = q_2$, and $y = q_3$) and the fact that $q_2 \geq q_3 \geq 1$ (implied by Lemma 1), and the second step holds by Fact 1(b) (with $b = \beta$, $x = V_{h+1}^k(s)$, and $y = V_{h+1}^\pi(s)$) and $H \geq V_{h+1}^k(s) \geq V_{h+1}^\pi(s) \geq 0$.

**Case $\beta < 0$.** The assumption $V_{h+1}^k(s') \geq V_{h+1}^\pi(s')$ for all $s' \in \mathcal{S}$ implies that $q_2 \leq q_3$ and therefore $G_2 \geq 0$ due to its definition (18). We also have
$$G_2 = \frac{1}{(-\beta)}\left(\log\{q_3\} - \log\{q_2\}\right)$$
$$\leq \frac{e^{-\beta H}}{(-\beta)}(q_3 - q_2)$$
$$\leq e^{|\beta|H}\mathbb{E}_{s' \sim P_h(\cdot \mid s, a)}[V_{h+1}^k(s') - V_{h+1}^\pi(s')],$$
where the second step holds by Fact 1(a) (with $g_0 = e^{\beta H}$, $x = q_3$, and $y = q_2$) and the fact that $q_3 \geq q_2 \geq e^{\beta H}$ (suggested by Lemma 1), and the third step holds by Fact 1(b) (with $b = \beta$, $x = V_{h+1}^k(s)$, and $y = V_{h+1}^\pi(s)$) and $V_{h+1}^k(s) \geq V_{h+1}^\pi(s) \geq 0$. $\square$

With the help of Lemma (2), we can show the "optimism" of $Q_h^k$ in the following sense.

**Lemma 3** *Suppose (19) holds with $G_1 \geq 0$. We have $Q_h^k(s, a) \geq Q_h^\pi(s, a)$ for all $(k, h, s, a) \in [K] \times [H] \times \mathcal{S} \times \mathcal{A}$.*

*Proof.* For the purpose of the proof, we set $Q_{H+1}^\pi(s, a) = Q_{H+1}^*(s, a) = 0$ for all $(s, a) \in \mathcal{S} \times \mathcal{A}$. We fix a tuple $(k, s, a) \in [K] \times \mathcal{S} \times \mathcal{A}$ and use strong induction on $h$. The base case for $h = H + 1$ is satisfied since $(Q_{H+1}^k - Q_{H+1}^\pi)(s, a) = 0$ for $k \in [K]$ by definition. Now we fix an $h \in [H]$ and assume that $0 \leq (Q_{h+1}^k - Q_{h+1}^\pi)(s, a)$. By the induction assumption we have
$$V_{h+1}^k(s) = \max_{a' \in \mathcal{A}} Q_{h+1}^k(s, a') \geq \max_{a' \in \mathcal{A}} Q_{h+1}^\pi(s, a') \geq V_{h+1}^\pi(s). \tag{20}$$

Applying (20) to Lemma 2 yields $G_2 \geq 0$. Since $G_1 \geq 0$ by assumption, it follows that $(Q_h^k - Q_h^\pi)(s, a) \geq 0$ by (19). The induction is completed and so is the proof. $\square$

Lemma 3 implies an immediate but important corollary.

**Lemma 4** *Suppose* (19) *holds with* $G_1 \geq 0$. *We have* $V_h^k(s) \geq V_h^\pi(s)$ *for all* $(k, h, s) \in [K] \times [H] \times \mathcal{S}$.

*Proof.* The result follows from Lemma 3 and Equation (20). $\qquad\square$

We now have all the keys to proving the meta regret bound.

## D  PROOF OF THEOREM 1

We work on the event of Condition 2 (which is a restatement of Condition 1), where $g$ and $\{m_h^k\}$ are defined. This means (19) also holds. Define $\delta_h^k := V_h^k(s_h^k) - V_h^{\pi^k}(s_h^k)$, and $\zeta_{h+1}^k := \mathbb{E}_{s' \sim P_h(\cdot \mid s_h^k, a_h^k)}[V_{h+1}^k(s') - V_{h+1}^{\pi^k}(s')] - \delta_{h+1}^k$. Let $\{m_h^k\}$ be as defined in Condition 2. For any $(k, h) \in [K] \times [H]$, we have

$$
\begin{aligned}
\delta_h^k &= (Q_h^k - Q_h^{\pi^k})(s_h^k, a_h^k) \\
&\leq \min\{H, (Q_h^k - Q_h^{\pi^k})(s_h^k, a_h^k)\} \\
&\leq \min\left\{H, c_1 \cdot \frac{e^{|\beta|H} - 1}{|\beta|} \cdot g \cdot m_h^k(s_h^k, a_h^k)\right\} + e^{|\beta|H} \cdot \mathbb{E}_{s' \sim P_h(\cdot \mid s_h^k, a_h^k)}[V_{h+1}^k(s') - V_{h+1}^{\pi^k}(s')] \\
&\leq c_1 \cdot \frac{e^{|\beta|H} - 1}{|\beta|} \cdot g \cdot \min\{1, m_h^k(s_h^k, a_h^k)\} + e^{|\beta|H} \cdot \mathbb{E}_{s' \sim P_h(\cdot \mid s_h^k, a_h^k)}[V_{h+1}^k(s') - V_{h+1}^{\pi^k}(s')] \\
&= c_1 \cdot \frac{e^{|\beta|H} - 1}{|\beta|} \cdot g \cdot \min\{1, m_h^k(s_h^k, a_h^k)\} + e^{|\beta|H}(\delta_{h+1}^k + \zeta_{h+1}^k).
\end{aligned}
$$

(21)

(22)

(23)

In the above equation, the first step holds by the construction of Algorithm 1 and the definition of $V_h^{\pi^k}$ in (4); the second step is due to the fact that $Q_h^k(\cdot, \cdot) \leq H$ and $Q_h^{\pi^k}(\cdot, \cdot) \geq 0$; the third step holds by (19) combined with Condition 2 and Lemma 2; the fourth step holds since $c_1, g \geq 1$ and $\frac{e^{|\beta|H} - 1}{|\beta|} \geq H$; the last step follows from the definitions of $\delta_h^k$ and $\zeta_{h+1}^k$.

Recalling from Algorithm 1 and the Bellman equation (4) that $V_{H+1}^k(s) = V_{H+1}^{\pi^k}(s) = 0$, as well as noting the fact that $\delta_{h+1}^k + \zeta_{h+1}^k \geq 0$ implied by Lemma 4, we can continue by expanding the recursion in Equation (23) and get

$$
\delta_1^k \leq \sum_{h \in [H]} e^{|\beta|Hh}\zeta_{h+1}^k + c_1 \cdot \frac{e^{|\beta|H} - 1}{|\beta|} \cdot \sum_{h \in [H]} e^{|\beta|H(h-1)}g \cdot \min\{1, m_h^k(s_h^k, a_h^k)\}. \tag{24}
$$

Therefore, we have

$$
\begin{aligned}
\text{Regret}(K) &= \sum_{k \in [K]} \left[(V_1^* - V_1^{\pi^k})(s_1^k)\right] \leq \sum_{k \in [K]} \delta_1^k \\
&\leq e^{|\beta|H^2} \sum_{k \in [K]} \sum_{h \in [H]} \zeta_{h+1}^k + c_1 \cdot \frac{e^{|\beta|H} - 1}{|\beta|} \cdot e^{|\beta|H^2} \cdot g \sum_{k \in [K]} \sum_{h \in [H]} \min\{1, m_h^k(s_h^k, a_h^k)\},
\end{aligned}
$$

(25)

where the second step holds by Lemma 4 with $\pi$ therein set to the optimal policy, and in the last step we have applied (24) along with the Holder inequality.

We proceed to control the first term in Equation (25). Since the construction of $V_h^k$ is independent of the new observation $s_h^k$ in episode $k$, we have that $\{\zeta_{h+1}^k\}$ is a martingale difference sequence satisfying $|\zeta_h^k| \leq 2H$ for all $(k, h) \in [K] \times [H]$. By the Azuma-Hoeffding inequality, we have for any $t > 0$,

$$
\mathbb{P}\left(\sum_{k \in [K]} \sum_{h \in [H]} \zeta_{h+1}^k \geq t\right) \leq \exp\left(-\frac{t^2}{2T \cdot H^2}\right).
$$

Hence, with probability $1 - \delta/2$, there holds

$$\sum_{k \in [K]} \sum_{h \in [H]} \zeta_{h+1}^k \leq \sqrt{2H^2 T \cdot \log(2/\delta)}. \tag{26}$$

Finally, plugging (26) into (25) yields

$$\text{Regret}(K) \leq e^{|\beta| H^2} \sqrt{2H^2 T \cdot \log(2/\delta)} + c_1 \cdot \frac{e^{|\beta| H} - 1}{|\beta|} \cdot e^{|\beta| H^2} \cdot g \sum_{k \in [K]} \sum_{h \in [H]} \min\{1, m_h^k(s_h^k, a_h^k)\}.$$

We then rescale $\delta$ properly and finish the proof of Theorem 1.

To obtain regret bounds for Algorithms 2 and 3, it remains to only specify quantities $g$ and $\sum_{k \in [K]} \sum_{h \in [H]} m_h^k(s_h^k, a_h^k)$ in Condition 2. The next result prepares us for the quest.

**Lemma 5** *Let* $q_2 = q_h^k(s, a)$ *be as defined in* (16). *If for each* $(k, h, s, a) \in [K] \times [H] \times \mathcal{S} \times \mathcal{A}$, *there exists some quantities* $g \geq 1$, $m_h^k = m_h^k(s, a) > 0$, $q_1 = q_{h,1}^k(s, a) \geq \min\{1, e^{\beta(H-h)}\}$ *and universal constant* $c_1 \geq 1$ *such that* $G_1(q_{h,1}^k(s, a)) = Q_h^k(s, a) - \frac{1}{\beta} \log\{q_{h,2}^k(s, a)\}$, *and* $0 \leq q_1 - q_2 \leq c_1 |e^{\beta H} - 1| g \cdot m_h^k$ *for* $\beta > 0$ *and* $0 \leq q_2 - q_1 \leq c_1 |e^{\beta H} - 1| g \cdot m_h^k$ *for* $\beta < 0$, *then Condition* 2 *holds with the aforementioned* $c_1$, $g$, $m_h^k$ *and* $q_1$.

*Proof.* **Case** $\beta > 0$. By the definition of $G_1$ in (19), the assumption $0 \leq q_1 - q_2$ implies that $G_1 \geq 0$. Moreover, Lemma 1 and Fact 1(a) (with $g_0 = 1$, $x = q_1$ and $y = q_2$) together imply

$$G_1 \leq \frac{1}{\beta}(q_1 - q_2).$$

Invoking the upper bound on $q_1 - q_2$ for $\beta > 0$ in the assumption completes the proof for the case.

**Case** $\beta < 0$. By the definition of $G_1$ in 18, the assumption $0 \leq q_2 - q_1$ implies that $G_1 \geq 0$. Furthermore, by Lemma 1 and Fact 1(a) (with $g_0 = e^{\beta H}$, $x = q_2$ and $y = q_1$), we further have

$$G_1 = \frac{1}{(-\beta)} \left(\log\{q_2\} - \log\{q_1\}\right)$$

$$\leq \frac{e^{-\beta H}}{|\beta|}(q_2 - q_1).$$

Invoking the upper bound on $q_2 - q_1$ and the fact that $|e^{\beta H} - 1| = 1 - e^{\beta H}$ for $\beta < 0$ completes the proof for the case. $\qquad \square$

## E   PROOF OF THEOREM 2

First of all, it can be seen that Line 7 in Algorithm 2 and the initial condition $V_{H+1}^k(s, a) = 0$ in Algorithm 1 ensure that $Q_h^k(s, a) \in [0, H - h + 1]$ for all $(k, h, s, a) \in [K] \times [H] \times \mathcal{S} \times \mathcal{A}$. We let $\lambda = (e^{\beta H} - 1)^2$ and $\gamma_L$ set as in (9) for Algorithm 2. We define

$$q_1^+ = q_{h,1}^{k,+}(s, a) := \begin{cases} \langle \phi_h^k(s, a), w_h^k \rangle + 1 + b_h^k(s, a), & \text{if } \beta > 0, \\ \langle \phi_h^k(s, a), w_h^k \rangle + 1 - b_h^k(s, a), & \text{if } \beta < 0; \end{cases} \tag{27}$$

$$q_1 = q_{h,1}^k(s, a) := \begin{cases} \min\{e^{\beta(H-h)}, q_1^+\}, & \text{if } \beta > 0, \\ \max\{e^{\beta(H-h)}, q_1^+\}, & \text{if } \beta < 0. \end{cases} \tag{28}$$

Indeed, $q_1$ defined above is equivalent to $q_{h,L}^k$ defined in (8). It can also be verified that $G_1(q_1) = Q_h^k(s, a) - \frac{1}{\beta} \log\{q_2\}$, where $G_1(\cdot)$ is defined in (18) and $q_2$ is defined in (16). We have the following result which shows that Algorithm 2 satisfies Condition 2 (a restatement of Condition 1) with high probability.

**Lemma 6** *Under Assumption 1, for any $\delta \in (0, 1]$, with probability at least $1 - \delta$, Condition 2 holds for Algorithm 2 with $c_1 \geq 1$, $g = \sqrt{d \log(2dKH/\delta)}$. $m_h^k(s, a) = \sqrt{\phi_h^k(s, a)^\top (\Lambda_h^k)^{-1} \phi_h^k(s, a)}$ and $q_1$ as defined in (28).*

*Proof.* Let us fix a tuple $(k, h, s, a) \in [K] \times [H] \times \mathcal{S} \times \mathcal{A}$. Then, we have

$$\phi_h^k(s, a)^\top w_h^k + 1 = \phi_h^k(s, a)^\top (\Lambda_h^k)^{-1} \left[ \sum_{\tau \in [k-1]} \phi_h^\tau(s_h^\tau, a_h^\tau) \cdot [e^{\beta \cdot V_{h+1}^\tau(s_{h+1}^\tau)} - 1] \right] + 1 \quad (29)$$

by Line 4 of Algorithm 2, and

$$
\begin{aligned}
\mathbb{E}_{s' \sim P_h(\cdot \mid s, a)} e^{\beta \cdot V_{h+1}^k(s')} &= \int_{\mathcal{S}} \psi(s, a, s')^\top \theta_h (e^{\beta \cdot V_{h+1}^k(s')} - 1) \mathrm{d}s' + 1 \\
&= \phi_h^k(s, a)^\top \theta_h + 1 \\
&= \phi_h^k(s, a)^\top (\Lambda_h^k)^{-1} \Lambda_h^k \theta_h + 1 \\
&= \phi_h^k(s, a)^\top (\Lambda_h^k)^{-1} \left[ \sum_{\tau \in [k-1]} \phi_h^\tau(s_h^\tau, a_h^\tau) \phi_h^\tau(s_h^\tau, a_h^\tau)^\top \theta_h + \lambda \cdot \theta_h \right] + 1 \\
&= \phi_h^k(s, a)^\top (\Lambda_h^k)^{-1} \left[ \sum_{\tau \in [k-1]} \phi_h^\tau(s_h^\tau, a_h^\tau) \cdot \mathbb{E}_{s' \sim P_h(\cdot \mid s_h^\tau, a_h^\tau)} [e^{\beta \cdot V_{h+1}^\tau(s')} - 1] + \lambda \cdot \theta_h \right] + 1,
\end{aligned}
$$

$$(30)$$

where the first step follows from Assumption 1, the second step holds by Line 5 of Algorithm 2, the third step holds since $\Lambda_h^k$ is positive definite by construction, the fourth step holds by Line 3 of Algorithm 2, and the last step holds since

$$\phi_h^\tau(s, a)^\top \theta_h = \int_{\mathcal{S}} \psi(s, a, s')^\top \theta_h \cdot (e^{\beta \cdot V_{h+1}^\tau(s')} - 1) \mathrm{d}s' = \mathbb{E}_{s' \sim P_h(\cdot \mid s, a)} [e^{\beta \cdot V_{h+1}^\tau(s')} - 1]$$

for $\tau \in [K]$, which is due to Assumption 1. Now we consider the cases $\beta > 0$ and $\beta < 0$ separately.

**Case $\beta > 0$.** Recall $q_1^+$ defined in (27) and $q_2$ in (16). To control $G_1$, we can compute

$$
\begin{aligned}
&\left| q_1^+ - q_2 - b_h^k(s, a) \right| \\
&= \left| \phi_h^k(s, a)^\top w_h^k + 1 - \mathbb{E}_{s' \sim P_h(\cdot \mid s, a)} e^{\beta \cdot V_{h+1}^k(s')} \right| \\
&= \left| \underbrace{\phi_h^k(s, a)^\top (\Lambda_h^k)^{-1} \left[ \sum_{\tau \in [k-1]} \phi_h^\tau(s_h^\tau, a_h^\tau) \cdot \left( e^{\beta \cdot V_{h+1}^\tau(s_{h+1}^\tau)} - \mathbb{E}_{s' \sim P_h(\cdot \mid s_h^\tau, a_h^\tau)} [e^{\beta \cdot V_{h+1}^\tau(s')}] \right) \right]}_{S_1} \right. \\
&\qquad \left. - \underbrace{\lambda \cdot \phi_h^k(s, a)^\top (\Lambda_h^k)^{-1} \theta_h}_{S_2} \right| \leq |S_1| + |S_2|, \quad (31)
\end{aligned}
$$

where the first step holds by the definitions of $q_1^+$ and $q_2$, and the second step is implied by (29) and (30). We control each of $S_1$ and $S_2$. For $S_1$, we have

$$|S_1| \leq \left\| \sum_{\tau \in [k-1]} \phi_h^\tau(s_h^\tau, a_h^\tau) \cdot \left( e^{\beta \cdot V_{h+1}^\tau(s_{h+1}^\tau)} - \mathbb{E}_{s' \sim P_h(\cdot \mid s_h^\tau, a_h^\tau)} e^{\beta \cdot V_{h+1}^\tau(s')} \right) \right\|_{(\Lambda_h^k)^{-1}} \|\phi_h^k(s, a)\|_{(\Lambda_h^k)^{-1}}$$

by the Cauchy-Schwarz inequality. On the event of Lemma 8, we further have

$$|S_1| \leq c \left| e^{\beta H} - 1 \right| \sqrt{d \log(2dKH/\delta)} \cdot \|\phi_h^k(s, a)\|_{(\Lambda_h^k)^{-1}}$$

for some universal constant $c > 0$. Now for $S_2$, we have

$$|S_2| \leq \lambda \cdot \|\phi_h^k(s, a)\|_{(\Lambda_h^k)^{-1}} \cdot \|\theta_h\|_{(\Lambda_h^k)^{-1}}$$

$$\leq \sqrt{\lambda} \cdot \|\phi_h^k(s,a)\|_{(\Lambda_h^k)^{-1}} \cdot \|\theta_h\|_2$$
$$\leq \sqrt{\lambda d} \cdot \|\phi_h^k(s,a)\|_{(\Lambda_h^k)^{-1}},$$

where the first step holds by the Cauchy-Schwarz inequality, the second step holds since $\Lambda_h^k \succeq \lambda \cdot I_d$, and the last step holds by Assumption 1 that $\|\theta_h\|_2 \leq \sqrt{d}$. Plugging the bounds on $S_1$ and $S_2$ into (31), and using the fact that $\lambda = (e^{\beta H} - 1)^2$ and the definition of $b_h^k$, we have

$$\left| \phi_h^\tau(s,a)^\top w_h^k + 1 - \mathbb{E}_{s' \sim P_h(\cdot \mid s,a)} e^{\beta \cdot V_{h+1}^k} \right| \leq b_h^k(s,a).$$

We choose $c_\gamma = c + 1$ in the definition of $b_h^k(s,a)$ in Line 6 of Algorithm 2, and we have

$$0 \leq q_1^+ - q_2 \leq \underbrace{2c_\gamma \cdot \left|e^{\beta H} - 1\right|}_{c_1} \underbrace{\sqrt{d \log(2dKH/\delta)}}_{g} \cdot \underbrace{\|\phi_h^k(s,a)\|_{(\Lambda_h^k)^{-1}}}_{m_h^k(s,a)}. \tag{32}$$

Since (32) implies $q_1^+ \geq q_2$ and Lemma 1 implies $q_2 \leq e^{\beta(H-h)}$, we can infer that $q_1 \geq q_2$ from the definition of $q_1 = \min\{e^{\beta(H-h)}, q_1^+\}$ in (28). Then we have $0 \leq q_1 - q_2 \leq q_1^+ - q_2$. By Lemma 1, we also have $q_1 \geq q_2 \geq 1$.

**Case $\beta < 0$.** Similar to the case of $\beta > 0$, we have

$$\left| q_2 - q_1^+ - b_h^k(s,a) \right| \leq c \cdot \left|e^{\beta H} - 1\right| \sqrt{d \log(2dKH/\delta)} \cdot \|\phi_h^k(s,a)\|_{(\Lambda_h^k)^{-1}}.$$

If we choose $c_\gamma = c + 1$ in the definition of $b_h^k(s,a)$ in Line 6 of Algorithm 2, the above equation implies

$$0 \leq q_2 - q_1^+ \leq \underbrace{2c_\gamma \cdot \left|e^{\beta H} - 1\right|}_{c_1} \underbrace{\sqrt{d \log(2dKH/\delta)}}_{g} \cdot \underbrace{\|\phi_h^k(s,a)\|_{(\Lambda_h^k)^{-1}}}_{m_h^k(s,a)}. \tag{33}$$

Therefore, using the same reasoning as in the case of $\beta > 0$, we have $0 \leq q_2 - q_1 \leq q_2 - q_1^+$. Also, by the definition of $q_1$ in (28), we have $q_1 \geq e^{\beta(H-h)}$.

The proof is completed by invoking Lemma 5 and recalling the identity $\|\phi_h^k(s,a)\|_{(\Lambda_h^k)^{-1}} = \sqrt{\phi_h^k(s,a)^\top (\Lambda_h^k)^{-1} \phi_h^k(s,a)}$. $\qquad\square$

Next, we give a bound for the quantity $\sum_{k \in [K]} \sum_{h \in [H]} \min\{1, m_h^k(s_h^k, a_h^k)\}$.

**Lemma 7** *Under Assumption 1, let $\{m_h^k(s,a)\}$ be as defined in Lemma 6 and we have*

$$\sum_{k \in [K]} \sum_{h \in [H]} \min\{1, m_h^k(s_h^k, a_h^k)\} \leq \sqrt{2dKH^2 \iota},$$

*where $\iota = \log(2dK/\delta)$.*

*Proof.* We have

$$\sum_{k \in [K]} \sum_{h \in [H]} \min\{1, m_h^k(s_h^k, a_h^k)\}$$
$$\leq \sum_{h \in [H]} \sqrt{K} \sqrt{\sum_{k \in [K]} \min\{1, \phi_h^k(s_h^k, a_h^k)^\top (\Lambda_h^k)^{-1} \phi_h^k(s_h^k, a_h^k)\}}$$
$$\leq H\sqrt{2dK\iota}.$$

where the first step holds by the Cauchy-Schwarz inequality, and the last step holds by Lemma 10. $\square$

Recall the definition of $M$ from Theorem 1, and now its upper bound can be determined by combining Lemmas 6 and 7. Therefore, the proof of the theorem is completed.

### E.1 Auxiliary Lemmas

We first present a concentration result.

**Lemma 8** *Let $\lambda = (e^{\beta H} - 1)^2$ in Algorithm 2. There exists a universal constant $c > 0$ such that for any $\delta \in (0, 1]$ and $(k, h) \in [K] \times [H]$, with probability $1 - \delta$, we have*

$$\Big\| \sum_{\tau \in [k-1]} \phi_h^\tau(s_h^\tau, a_h^\tau) \cdot (e^{\beta \cdot V_{h+1}^\tau(s_{h+1}^\tau)} - \mathbb{E}_{s' \sim P_h(\cdot \mid s_h^\tau, a_h^\tau)} e^{\beta \cdot V_{h+1}^\tau(s')}) \Big\|_{(\Lambda_h^k)^{-1}} \leq c \left| e^{\beta H} - 1 \right| \sqrt{d \log(2dKH/\delta)}$$

*Proof.* The proof is adapted from that of Cai et al. (2019, Lemma D.1) by using the fact that for any function $V : \mathcal{S} \to [0, H]$, we have

$$\sup_{(s,a,s'') \in \mathcal{S} \times \mathcal{A} \times \mathcal{S}} \left| e^{\beta \cdot V(s'')} - \mathbb{E}_{s' \sim P_h(\cdot \mid s,a)} e^{\beta \cdot V(s')} \right| \leq \left| e^{\beta H} - 1 \right|,$$

and that Assumption 1 implies

$$\sup_{(s,a) \in \mathcal{S} \times \mathcal{A}} \Big\| \int_{\mathcal{S}} \phi(s, a, s') \cdot (e^{\beta \cdot V(s')} - 1) \mathrm{d}s' \Big\|_2 \leq \sqrt{d} \left| e^{\beta H} - 1 \right|.$$

$\square$

The next few lemmas can help control the sum of the terms $\{\phi_h^k(s_h^k, a_h^k)^\top (\Lambda_h^k)^{-1} \phi_h^k(s_h^k, a_h^k)\}$.

**Lemma 9 (Cai et al. (2019, Lemma D.3))** *Let $\{\phi_j\}_{j \geq 1}$ be a sequence in $\mathbb{R}^d$. Let $\Lambda_0 \in \mathbb{R}^{d \times d}$ be a positive-definite matrix and $\Lambda_t := \Lambda_0 + \sum_{j \in [t-1]} \phi_j \phi_j^\top$. Then for any $t \in \mathbb{Z}_{>0}$, we have*

$$\sum_{j \in [t]} \min\{1, \phi_j^\top \Lambda_j^{-1} \phi_j\} \leq 2 \log \left[ \frac{\det(\Lambda_{t+1})}{\det(\Lambda_1)} \right].$$

**Lemma 10** *Let $\lambda = (e^{\beta H} - 1)^2$ in Algorithm 2. For any $h \in [H]$, we have*

$$\sum_{k \in [K]} \min\{1, \phi_h^k(s_h^k, a_h^k)^\top (\Lambda_h^k)^{-1} \phi_h^k(s_h^k, a_h^k)\} \leq 2d\iota,$$

*where $\iota = \log(2dK/\delta)$.*

*Proof.* By construction of Algorithm 2, we may define $\Lambda_h^0 := \lambda I_d$ so we have

$$\Lambda_h^k = \Lambda_h^0 + \sum_{\tau \in [k-1]} \phi_h^\tau(s_h^\tau, a_h^\tau) \phi_h^\tau(s_h^\tau, a_h^\tau)^\top.$$

Since $\|\phi_h^k(s_h^k, a_h^k)\|_2 \leq \sqrt{d} \left| e^{\beta H} - 1 \right|$ for all $(k, h) \in [K] \times [H]$ as implied by Assumption 1, we have for any $h \in [H]$ that

$$\Lambda_h^{K+1} = \sum_{k \in [K]} \phi_h^k(s_h^k, a_h^k) \phi_h^k(s_h^k, a_h^k)^\top + \lambda I_d \preceq (dK|e^{\beta H} - 1|^2 + \lambda) I_d.$$

Given $\lambda = \left| e^{\beta H} - 1 \right|^2$, we have for any $h \in [H]$ that

$$\log \left[ \frac{\det(\Lambda_h^{K+1})}{\det(\Lambda_h^1)} \right] \leq d \log \left[ \frac{dK \left| e^{\beta H} - 1 \right|^2 + \lambda}{\lambda} \right]$$
$$\leq d \log[dK + 1]$$
$$\leq d\iota.$$

We now apply Lemma 9 to get

$$\sum_{k \in [K]} \min\{1, \phi_h^k(s_h^k, a_h^k)^\top (\Lambda_h^k)^{-1} \phi_h^k(s_h^k, a_h^k)\} \leq 2 \log \left[ \frac{\det(\Lambda_h^{K+1})}{\det(\Lambda_h^1)} \right]$$
$$\leq 2d\iota,$$

as desired. $\square$

## F    PROOF OF COROLLARY 1

The result follows from Theorem 2, as well as the fact that $\lim_{\beta \to 0} \frac{e^{|\beta|H} - 1}{|\beta|} = H$ and $\lim_{\beta \to 0} e^{|\beta|H^2} = 1$.

## G    PROOF OF THEOREM 3

First of all, it can be seen that Line 5 in Algorithm 3 and the initial condition $V_{H+1}^k(s,a) = 0$ in Algorithm 1 ensure that $Q_h^k(s,a) \in [0, H-h+1]$ for all $(k,h,s,a) \in [K] \times [H] \times \mathcal{S} \times \mathcal{A}$. We fix a tuple $(k,h,s,a) \in [K] \times [H] \times \mathcal{S} \times \mathcal{A}$. Recall that $\gamma_G$ is as defined in (13). For Algorithm 3, we let

$$q_1 = q_{h,1}^k(s,a) := \begin{cases} \max_{P \in \mathcal{P}_h^k} \int_{\mathcal{S}} P(s' \mid s,a) e^{\beta \cdot V_{h+1}^k(s')} \mathrm{d}s', & \text{if } \beta > 0, \\ \min_{P \in \mathcal{P}_h^k} \int_{\mathcal{S}} P(s' \mid s,a) e^{\beta \cdot V_{h+1}^k(s')} \mathrm{d}s', & \text{if } \beta < 0, \end{cases} \tag{34}$$

which is equivalent to $q_{h,G}^k(s,a)$ defined in (15). Recall the definitions of $z_p$ and $\mathcal{Z}$ in (11). We have the result below, which verifies Condition 2 (a restatement of Condition 1) for Algorithm 3 under the general function approximation.

**Lemma 11** *Under Assumption 2, for any $\delta \in (0,1]$ the following holds with probability at least $1-\delta$. For all $(k,h,s,a) \in [K] \times [H] \times \mathcal{S} \times \mathcal{A}$, Condition 2 holds for Algorithm 3 with $c_1 = 1$, $g = 1$, $m_h^k(s,a) = \frac{1}{|e^{\beta H} - 1|} \left[ \max_{P \in \mathcal{P}_h^k} z_P(s,a,V_{h+1}^k) - \min_{P \in \mathcal{P}_h^k} z_P(s,a,V_{h+1}^k) \right]$ and $q_1$ as defined in (34).*

*Proof.* It is not hard to see that by the definition of $q_1$ in (34), we have

$$q_1 \in [\min\{1, e^{\beta(H-h)}\}, \max\{1, e^{\beta(H-h)}\}].$$

Recall the definitions of $q_2$ in (16) and $G_1(\cdot)$ in (18); it holds that $G_1(q_1) = Q_h^k - \frac{1}{\beta} \log\{q_2\}$. On the event of Lemma 14, we have $P_h \in \mathcal{P}_h^k$.

**Case $\beta > 0$.** By definitions of $q_1$ and $q_2$ and the fact that $P_h \in \mathcal{P}_h^k$, we have $q_1 \geq q_2$. We can also derive

$$
\begin{aligned}
q_1 - q_2 &= \max_{P \in \mathcal{P}_h^k} \int_{\mathcal{S}} P(s' \mid s,a) e^{\beta \cdot V_{h+1}^k(s')} \mathrm{d}s' - \int_{\mathcal{S}} P_h(s' \mid s,a) e^{\beta \cdot V_{h+1}^k(s')} \mathrm{d}s' \\
&\leq \max_{P \in \mathcal{P}_h^k} \int_{\mathcal{S}} P(s' \mid s,a) e^{\beta \cdot V_{h+1}^k(s')} \mathrm{d}s' - \min_{P \in \mathcal{P}_h^k} \int_{\mathcal{S}} P(s' \mid s,a) e^{\beta \cdot V_{h+1}^k(s')} \mathrm{d}s' \\
&= \max_{P \in \mathcal{P}_h^k} \int_{\mathcal{S}} P(s' \mid s,a)(e^{\beta \cdot V_{h+1}^k(s')} - 1) \mathrm{d}s' - \min_{P \in \mathcal{P}_h^k} \int_{\mathcal{S}} P(s' \mid s,a)(e^{\beta \cdot V_{h+1}^k(s')} - 1) \mathrm{d}s' \\
&= |e^{\beta H} - 1| g \cdot m_h^k(s,a),
\end{aligned}
$$

where the second step holds since $P_h \in \mathcal{P}_h^k$, and the third step holds since $\int_{\mathcal{S}} P(s' \mid s,a) \mathrm{d}s' = 1$.

**Case $\beta < 0$.** By definitions of $q_1$ and $q_2$ and the fact that $P_h \in \mathcal{P}_h^k$, we may deduce that $q_1 \leq q_2$. Also, we have

$$
\begin{aligned}
q_2 - q_1 &= \int_{\mathcal{S}} P_h(s' \mid s,a) e^{\beta \cdot V_{h+1}^k(s')} \mathrm{d}s' - \min_{P \in \mathcal{P}_h^k} \int_{\mathcal{S}} P(s' \mid s,a) e^{\beta \cdot V_{h+1}^k(s')} \mathrm{d}s' \\
&\leq \max_{P \in \mathcal{P}_h^k} \int_{\mathcal{S}} P(s' \mid s,a) e^{\beta \cdot V_{h+1}^k(s')} \mathrm{d}s' - \min_{P \in \mathcal{P}_h^k} \int_{\mathcal{S}} P(s' \mid s,a) e^{\beta \cdot V_{h+1}^k(s')} \mathrm{d}s' \\
&= -\min_{P \in \mathcal{P}_h^k} \int_{\mathcal{S}} P(s' \mid s,a)(-e^{\beta \cdot V_{h+1}^k(s')}) \mathrm{d}s' - (-1) \max_{P \in \mathcal{P}_h^k} \int_{\mathcal{S}} P(s' \mid s,a)(-e^{\beta \cdot V_{h+1}^k(s')}) \mathrm{d}s' \\
&= \max_{P \in \mathcal{P}_h^k} \int_{\mathcal{S}} P(s' \mid s,a)(1 - e^{\beta \cdot V_{h+1}^k(s')}) \mathrm{d}s' - \min_{P \in \mathcal{P}_h^k} \int_{\mathcal{S}} P(s' \mid s,a)(1 - e^{\beta \cdot V_{h+1}^k(s')}) \mathrm{d}s' \\
&= |e^{\beta H} - 1| g \cdot m_h^k(s,a),
\end{aligned}
$$

where the second step holds since $P_h \in \mathcal{P}_h^k$, and the fourth step holds since $\int_{\mathcal{S}} P(s' \mid s, a) \mathrm{d}s' = 1$. Finally, invoking Lemma 5 completes the proof. $\qquad\square$

The following lemma controls $\sum_{k \in [K]} \sum_{h \in [H]} \min\{1, m_h^k(s_h^k, a_h^k)\}$.

**Lemma 12** *Let $d := \dim_E(\mathcal{Z}, |e^{\beta H} - 1|/K)$. Under Assumption 2, for any $\delta \in (0, 1]$, let $\{m_h^k(s, a)\}$ be as defined in Lemma 11 and $\zeta$ be as defined in (12), then with probability at least $1 - \delta$, we have*

$$\sum_{k \in [K]} \sum_{h \in [H]} \min\{1, m_h^k(s_h^k, a_h^k)\} \le 2H \min\{d, K\} + 20H\sqrt{dK\zeta}.$$

*Proof.* Since $\min\{1, m_h^k(s_h^k, a_h^k)\} \le m_h^k(s_h^k, a_h^k)$ for all $(k, h) \in [K] \times [H]$. For each fixed $h \in [H]$, by Lemma 15, we have

$$\sum_{k \in [K]} \min\{1, m_h^k(s_h^k, a_h^k)\} \le \frac{1}{|e^{\beta H} - 1|} \left[ |e^{\beta H} - 1| + |e^{\beta H} - 1| \cdot \min\{d, K\} + 5\sqrt{\gamma^2 dK} \right]$$

$$\le \frac{1}{|e^{\beta H} - 1|} \left[ 2|e^{\beta H} - 1| \cdot \min\{d, K\} + 5\sqrt{\gamma^2 dK} \right]$$

$$= 2 \cdot \min\{d, K\} + \frac{5}{|e^{\beta H} - 1|}\sqrt{\gamma^2 dK}$$

$$\le 2 \cdot \min\{d, K\} + 20\sqrt{dK \left[ \log\left(\mathcal{N}_{1/K}(\mathcal{P}, \|\cdot\|_{\infty,1}) \cdot H/\delta\right) + \sqrt{\log(4K^2 H/\delta)} \right]},$$

where the last step holds since $\gamma_G = \gamma$ where $\gamma$ is given in Lemma 14. Summing both sides of the above equations over $h \in [H]$ results in the desired bound. $\qquad\square$

Recall the definition of $M$ from Theorem 1, and now its upper bound can be determined by combining Lemmas 11 and 12. Therefore, the proof of the theorem is completed.

### G.1 AUXILIARY LEMMAS

Let $\overline{\mathcal{Z}}$ be a set of $[0, D]$-valued functions for some number $D > 0$. We define $\{(X_\tau, Y_\tau)\}_{\tau \in [t]}$ be a series of random variables such that each $X_\tau$ is in the domain of the elements of function set $\overline{\mathcal{Z}}$, and each $Y_\tau \in \mathbb{R}$. Let $\mathcal{F} = \{\mathcal{F}_\tau\}_{\tau \ge 1}$ be a set of filtrations such that for all $\tau \ge 1$, the random variables $\{X_1, Y_1, \ldots, X_{\tau-1}, Y_{\tau-1}, X_\tau\}$ is $\mathcal{F}_{\tau-1}$-measurable. Furthermore, we assume there exists a function $z^* \in \overline{\mathcal{Z}}$ such that $\mathbb{E}[Y_\tau \mid \mathcal{F}_{\tau-1}] = z^*(X_\tau)$. For any $\varepsilon > 0$, we denote by $\mathcal{N}_\varepsilon(\overline{\mathcal{Z}}, \|\cdot\|_\infty)$ the $\varepsilon$-covering number of $\overline{\mathcal{Z}}$ with respect to the supremum norm $\|z_1 - z_2\|_\infty = \sup_x |z_1(x) - z_2(x)|$. We define

$$\hat{z}_t := \operatorname*{argmin}_{z \in \overline{\mathcal{Z}}} \sum_{\tau \in [t]} (z(X_\tau) - Y_\tau)^2,$$

and for $\gamma \ge 0$, let

$$\mathcal{Z}_t(\gamma) := \left\{ z \in \overline{\mathcal{Z}} : \sum_{\tau \in [t]} (z(X_\tau) - \hat{z}_t(X_\tau))^2 \le \gamma^2 \right\}.$$

We record a concentration result.

**Lemma 13** *Suppose that for any $\tau \ge 1$, the random variable $Y_\tau - z^*(X_\tau)$ is conditionally $\sigma$-sub-Gaussian given filtration $\mathcal{F}_{\tau-1}$. Let*

$$\gamma_t^2(\delta, \varepsilon) := 8\sigma^2 \log\left(\mathcal{N}_\varepsilon(\overline{\mathcal{Z}}, \|\cdot\|_\infty)/\delta\right) + 4\varepsilon t \left(D + \sqrt{\sigma^2 \log(4t(t+1)/\delta)}\right).$$

*Then for any $\varepsilon > 0$ and $\delta \in (0, 1]$, with probability at least $1 - \delta$ we have $z^* \in \mathcal{Z}_t(\gamma_t(\delta, \varepsilon))$.*

*Proof.* The proof can be adapted from that of Russo & Van Roy (2014, Proposition 6). □

In the following, we use the shorthand $\gamma := \gamma_G$, where $\gamma_G$ is defined in (13) and used in Line 4 of Algorithm 3.

**Lemma 14** *For any $\delta \in (0, 1]$ and*

$$\gamma^2 = 10 \left| e^{\beta H} - 1 \right|^2 \left[ \log \left( \mathcal{N}_{1/K}(\mathcal{P}, \| \cdot \|_{\infty,1}) \cdot H/\delta \right) + \sqrt{\log(4K^2 H/\delta)} \right],$$

*then for all $(k, h) \in [K] \times [H]$ we have $P_h \in \mathcal{P}_h^k$ with probability at least $1 - \delta$.*

*Proof.* We first note that for any $(k, h) \in [K] \times [H]$, Line 3 in Algorithm 3 can be equivalently written as

$$P_h^k \leftarrow \underset{P \in \mathcal{P}}{\text{argmin}} \sum_{\tau \in [k-1]} ((e^{\beta \cdot V_{h+1}^\tau(s_{h+1}^\tau)} - 1) - \int_{\mathcal{S}} P(s' \mid s_h^\tau, a_h^\tau) \cdot (e^{\beta \cdot V_{h+1}^\tau(s')} - 1) ds')^2,$$

since $\int_{\mathcal{S}} P(s' \mid s_h^\tau, a_h^\tau) ds' = 1$. Recall the definition of $z_P$ in (11). For any $(k, h) \in [K] \times [H]$, we set $\overline{\mathcal{Z}} = \mathcal{Z}$, $Y_k = e^{\beta \cdot V_{h+1}^k(s_{h+1}^k)} - 1$, $X_k = (s_h^k, a_h^k, V_{h+1}^k)$ and $z^* = z_{P_h}$. Then, we have that $Y_\tau - z^*(X_\tau)$ is conditionally $(\left| e^{\beta H} - 1 \right|)$-sub-Gaussian for $\tau \in [k-1]$ given a properly defined filtration. By the definition of $\mathcal{P}_h^k$, we have $\mathcal{Z}_k(\gamma) = \{z_P : P \in \mathcal{P}_h^k\}$. By definition of $\gamma$, we have

$$\gamma \geq \gamma_{k-1}(\delta/H, \left| e^{\beta H} - 1 \right|/K)$$

for all $k \in [K]$, where $\gamma_t(\cdot, \cdot)$ is as defined in Lemma 13. By Lemma 13 with $D = \sigma = \left| e^{\beta H} - 1 \right|$ and $\varepsilon = \left| e^{\beta H} - 1 \right|/K$, with probability at least $1 - \delta/H$ and for all $k \in [K]$, we have

$$z^* \in \mathcal{Z}_k(\gamma_{k-1}(\delta/H, \left| e^{\beta H} - 1 \right|/K)) \subset \mathcal{Z}_k(\gamma),$$

thus implying $P_h \in \mathcal{P}_h^k$. Applying the union bound over $h \in [H]$, we have that $P_h \in \mathcal{P}_h^k$ with probability at least $1 - \delta$. Now we show that $\mathcal{N}_\varepsilon(\mathcal{Z}, \| \cdot \|_\infty) \leq \mathcal{N}_{\varepsilon/|e^{\beta H}-1|}(\mathcal{P}, \| \cdot \|_{\infty,1})$ for any $\varepsilon > 0$. Let $\mathcal{V} := \{V : \mathcal{S} \to [0, H]\}$. For any $P, P' \in \mathcal{P}$ and their corresponding $z_P, z_{P'} \in \mathcal{Z}$, we can compute

$$\|z_P - z_{P'}\|_\infty = \sup_{(s,a,V) \in \mathcal{S} \times \mathcal{A} \times \mathcal{V}} \left| \int_{\mathcal{S}} P(s' \mid s, a)(e^{\beta \cdot V(s')} - 1) ds' - \int_{\mathcal{S}} P'(s' \mid s, a)(e^{\beta \cdot V(s')} - 1) ds' \right|$$

$$\leq \left| e^{\beta H} - 1 \right| \cdot \sup_{(s,a) \in \mathcal{S} \times \mathcal{A}} \int_{\mathcal{S}} |P(s' \mid s, a) - P'(s' \mid s, a)| \, ds'$$

$$= \left| e^{\beta H} - 1 \right| \cdot \|P - P'\|_{\infty,1},$$

as desired. □

We have the following result on the eluder dimension.

**Lemma 15** *Let $\overline{\mathcal{Z}} = \mathcal{Z}$ and $d = \dim_E(\mathcal{Z}, \left| e^{\beta H} - 1 \right|/K)$. For any $K \geq 1$, $\beta \in \mathbb{R}$ and $\bar{\gamma} > 0$, we have*

$$\sum_{k \in [K]} \sup_{z, z' \in \mathcal{Z}_k(\gamma)} |z(x_k) - z'(x_k)| \leq \left| e^{\beta H} - 1 \right| + \left| e^{\beta H} - 1 \right| \cdot \min\{d, K\} + 4\sqrt{\bar{\gamma}^2 dK}.$$

*Proof.* This result is an adaptation of Russo & Van Roy (2014, Lemma 5) with $C = \left| e^{\beta H} - 1 \right|$ therein. □

