# OpenReview forum: "Entropic Risk-Sensitive Reinforcement Learning: A Meta Regret Framework with Function Approximation"
_ICLR.cc/2021/Conference — Reject_

### Official Review · AnonReviewer4 · 2020-10-25
**Decent theoretical paper**

**Rating:** 6
**Confidence:** 3

**Review:**

##########################################################################

Summary:

The paper studies risk-sensitive reinforcement learning with the entropic risk measure and function approximation. A meta algorithm based on value iteration is first proposed, then the paper proposes two concrete instantiations, one for linear function approximation and one for general function approximation. Regret bound for both algorithms depend sub-linearly in the number of episodes, and the linear one depends polynomially on the ambient dimension, whereas the general one depends polynomially on the Eluder dimension.

##########################################################################

Reasons for score:

Overall, I vote for accepting. I think this paper considers an important topic: risk-sensitive reinforcement learning, and provides a first step in establishing theoretical foundation under that setting with function approximation.

##########################################################################

Pros:

1) The paper tackles an important issue: risk-sensitive reinforcement learning with function approximation, and shows regret bounds that are near optimal under some assumptions.

2) The value iteration based meta algorithm framework and the meta regret bound provides a nice abstraction beyond concrete implementation under different assumptions. It might benefit future studies on function approximation under risk-sensitive reinforcement learning.


##########################################################################

Cons:

1. The technical tools used in this paper seem to be based on a combination of ideas from Fei et al. (2020), Cai et al (2019), Russo and Van Roy (2014) and other papers. The analysis itself does not bring new insights.


##########################################################################

Minor comments:

It might be worthwhile to discuss other definition of “risks” beyond entropic risk measure under the risk-sensitive reinforcement learning settings, and mention any relevant theoretical results under those definitions.

---

> ### Author Response · Authors · 2020-11-24
> **Response to AnonReviewer4**
>
> Thank you for your positive feedback. We would like to illustrate the following new insights of our work compared to existing works.
>
> *) Linear function approximation:  please see our response to AnonReviewer1.
>
> *) General function approximation:   our Algorithm 3 is indeed inspired by [Russo and Van Roy (2014)]. However, naively following the analysis in [Russo and Van Roy (2014)]  would lead to the choice of function set $\cal{Z}$ in (11) with $e^{\beta \cdot V(s')} - 1$ replaced by  $e^{\beta \cdot V(s')}$, and doing so would add an extra multiplicative factor of $e^{\beta H} / \beta$ in our current regret bound when $\beta > 0$.
> We remark that this additional factor grows exponentially in $\beta$ and tends to infinity when $\beta \to 0$. As a consequence, one would end up with a bound that is *exponentially* worse than the current one, and would prevent one from recovering the standard bound in the risk-neutral setting as $\beta \to 0$.
> One of the key insights in the analysis of Algorithm 3 is that one should consider $\cal{Z}$ in (11) for function approximation, instead of the one suggested by the analysis in [Russo and Van Roy (2014)].

---

### Official Review · AnonReviewer2 · 2020-10-28
**not related to theme of the conference...**

**Rating:** 5
**Confidence:** 3

**Review:**

The paper considers a finite horizon episodic RL problem where the objective is risk-sensitive using the entropic risk measure. The paper proposes two algorithms based on the optimism in the face of uncertainty principle, one for linear function approximation and the other for general function approximation. Regret bounds are derived for both algorithms.

The strong points of this paper are that the algorithms proposed seem well grounded based on the optimism principle, and that a formal regret analysis is provided for general optimistic algorithms in the context of risk-sensitive object, which is then specialized to the two algorithms proposed.

One weak point of this paper is that the algorithms proposed do not seem very practical. What’s proposed for general function approximation is mostly a conceptual algorithm. It’s unclear how it could be applied in practice. Even with linear function approximation, it seems that the algorithm requires storing all past iterates as well as sweeping over all states and actions for each episode, which is extremely computationally costly.

Even though there’s some nice theoretical contributions, I am not sure if this is the best venue for this paper, since it’s not really related to representation learning… (although the conference has been constantly evolving over the years).

Given the points above, I am leaning towards rejection.

The organization of the paper is pretty clear. One minor suggestion is that the authors might want to compress some of the background on value functions to make room for a concluding section.

The paper is easy to read in general, but I find the naming and terminology like “meta RSVI” and “meta regret analysis” a bit confusing. It seems to me that the proposed algorithms fall straight under the OFU principle, and it might be more straightforward to just say that instead of coming up with new names like MetaRSVI.

---

> ### Author Response · Authors · 2020-11-24
> **Response to AnonReviewer2**
>
> We appreciate your review. Please see below for our response.
>
> *) "Practicality". Our work focuses on theoretical analysis, and our algorithms are designed for that purpose. That being said, it is still possible to run the algorithms in practical settings with reasonable computational complexity.
> In the setting of linear function approximation, one only needs to store the sampled trajectory of the current episode and previous episode, and the main step in Algorithm 2 (Line 4, which amounts to a least-squares update) can be implemented in an online fashion. In practice, various approximation techniques may be explored, including deployment of replay buffers, though this is beyond the scope of our work.
>
> *) "Relevance".
> Risk-sensitive RL is an important research area for RL applications where risk must be carefully handled, such as critical decision making processes involving human beings. We have listed various examples in the beginning of Section 1, and they are all practical applications of RL and AI. Hence, we think that our work is relevant to the theme of the conference.
>
> *) "Meta algorithm/regret". The meta algorithms/regret are used to disentangle  part of the analysis that relates to function approximation from the part that does not. We believe such formulation  is best for understanding the role of function approximation in regret analysis, as pointed out in Section 1 of our paper. This general framework is flexible and allows for analysis of other types of function approximation, in addition to linear and general function approximation studied in the paper.

---

### Official Review · AnonReviewer1 · 2020-10-29
**Entropic Risk-Sensitive Reinforcement Learning: A Meta Regret Framework with Function Approximation**

**Rating:** 4
**Confidence:** 4

**Review:**

This paper proposes a risk-sensitive algorithm with function approximation in reinforcement learning. To handle the uncertainty, the proposed algorithms consider an entropic risk value function controlled by a risk parameter, which provides a unified framework for both risk-sensitive and risk-averse settings. The main contribution of this paper is to provide theoretical guarantees for the proposed algorithms.

Pros

The idea of using entropic risk value functions in RL is a very interesting.

The paper is well-written and the proofs are present in details.

Cons

My major concern is the novelty in this paper. It is closely related to a recent paper, which considers the same entropic value function in RL, but focus on the tabular setting:

Fei, Y., Yang, Z., Chen, Y., Wang, Z. and Xie, Q., 2020. Risk-Sensitive Reinforcement Learning: Near-Optimal Risk-Sample Tradeoff in Regret. arXiv preprint arXiv:2006.13827.

This paper is trying to extend the results in [Fei et al. 2020] to the function approximation setting. Then the key to decide if the paper’s contribution is enough is to see how hard such extension is.

However, although [Fei et al. 2020] focus on the tabular setting, it seems that their proof techniques can be easily generalized to the linear function approximation setting (as they write the tabular representation using canonical basis, then the values could be written as the solution of a linear regression). I check the details of the proof and it indeed shares a lot of similarities with [Fei et al. 2020]. The major difference is how to choose a proper regularization (Lemma 10).

Minor comments:

Two related papers that use entropic value functions in RL.

O'Donoghue, B., 2018. Variational bayesian reinforcement learning with regret bounds. arXiv preprint arXiv:1807.09647.

O'Donoghue, B., Osband, I. and Ionescu, C., 2020. Making sense of reinforcement learning and probabilistic inference. arXiv preprint arXiv:2001.00805.

---

> ### Author Response · Authors · 2020-11-24
> **Response to AnonReviewer1**
>
> Thank you for your review. Our comments are as follows.
>
> *) "Comparison to [Fei et al (2020)]". The regularization is a critical part of our algorithm, but is not the only novelty of our Algorithm 2 compared to [Fei et al (2020)]. The need for regularization in linear function approximation makes algorithm design much more challenging.
> In particular, the value iteration algorithm in [Fei et al (2020)], even with regularization added and the canonical bases therein replaced by linear features, would incur an extra multiplicative factor of  $\log(\max(1,e^{\beta H}) / |e^{\beta H}-1|)$ in regret compared to our current bound. Note that this extra factor grows exponentially as  $\beta \to 0$. Therefore, when $\beta \to 0$, the resulting bound would be  *exponentially* worse than our current bound, and would prevent us from recovering the regret bound in the risk-neutral setting.
> The novelty of our Algorithm 2 lies in the fact that it overcomes the above issue by using regression target and regressors on the order of $e^{\beta H} - 1$ in the least-squares step (7) (or Line 4 in Algorithm 2).
> This distinctive feature of Algorithm 2 makes it significantly different from the value iteration algorithm proposed in [Fei et al (2020)], which is designed only for the tabular case.
>
> *) In addition to the linear function approximation, our paper also provides a risk-sensitive RL algorithm and regret bound for general function approximation. To the best of our knowledge, this is the first time that general function approximation is studied for risk-sensitive RL.

---

### Official Review · AnonReviewer3 · 2020-11-02
**Paper presents new theoretical (regret) results on entropic RL with linear parametrization, lack experiments**

**Rating:** 5
**Confidence:** 2

**Review:**

In this work the authors studied the entropic risk RL problem and derived a regret bound of the entropic value iteration algorithm via the risk-sensitive optimism in face of uncertainty approach. Utilizing the theoretical framework of Osband 2014, they extend the results to entropic-risk RL with linear function approximations on the transition dynamics and value function and later with general function approximation in episodic MDPs.

Unfortunately I am not very familiar with the recent theoretical exploration results with randomized value functions. On the high-level, I find the contribution of extending this to entropic-risk RL reasonable. One main question is why episodic MDPs are chosen for deriving the main theoretical results, and can these results be extended to more standard MDP frameworks such as discounting MDPs. Moreover, on top of the technical description, can the authors provide some intuitive explanations of condition 1 (which is a main RSOFU assumption that pivots most of the theoretical results) and some examples on when this condition is satisfied?

While I understand this paper's main contribution is theoretical and appreciate the efforts for deriving that. Since the algorithm in analysis is fairly standard, it'd be great to see at least some simple, proof-of-concept numerical experiments to evaluate the tightness of the regret bound. Without that, it's rather difficult to understand the several theoretical results, compare with the references mentioned, and justify their usefulness.

---

> ### Author Response · Authors · 2020-11-24
> **Response to AnonReviewer3**
>
> Thank you for your feedback. Please see our replies below.
>
> *) "Randomized value function". Our algorithms are quite different in nature from those based on randomized value function. Conditional on the sampled trajectory, our algorithms estimate parameters via deterministic update, whereas algorithms based on randomized value function sample parameter estimates from some Bayesian posterior distributions.
>
> *) "Episodic MDP".
> Episodic MDP is a standard setting for regret analysis of RL. Please see, e.g., [Azar et al (2017); Jin et al (2018); Yang and Wang (2019)] among many others. The  results in the episodic MDP can be extended to other MDP settings, such as discounted MDP by adapting techniques in [Zhou et al (2020)], though this is beyond the scope of this work.
>
> *) "Condition 1".
> The intuition of Condition 1 is that it ensures the stability of iterates {$\{Q^k_h\}$} while Algorithm 1 is running, so that {${Q^k_h}$} do not deviate too much from the ideal iterates {$\{\overline{Q}^k_h\}$} defined on the bottom of pp6.
> Condition 1 serves as an abstraction of optimism for proving the meta regret bound in Theorem 1. In the concrete results (Theorems 2 and 3) that follow the meta regret bound, we show that both Algorithms 2 and 3 (which are special instances of Algorithm 1)  satisfy Condition 1 with high probability, and therefore prove that they attain the claimed concrete regret bounds. We would like to emphasize that in Theorems 2 and 3, Condition 1 is *not* an assumption, but rather something that our algorithms are proved to satisfy.
>
> *) "Experiments".
> Our paper focuses on theoretical aspects of risk-sensitive RL. It is an excellent suggestion to conduct numerical experiments to support our theoretical results. We will work on this in the future.
>
> Reference:
>
> Azar et al (2017). Minimax Regret Bounds for Reinforcement Learning.
>
> Jin et al (2018). Is Q-learning Provably Efficient?
>
> Yang and Wang (2019). Reinforcement Learning in Feature Space: Matrix Bandit, Kernels, and Regret Bound.
>
> Zhou et al (2020). Provably Efficient Reinforcement Learning for Discounted MDPs with Feature Mapping.

---

### Decision · Program_Chairs · 2021-01-07
**Final Decision**

**Decision:**

Reject

**Comment:**


The paper considers the risk sensitive RL by exploiting entropic risk. The major contribution of this paper is providing the theoretical guarantees for the proposed risk-senstive value iteration with function approximation.

The major concern of this paper is the similarity to the existing work in (Fei et al., 2020). I encourage the authors to reorganize the paper and emphasize the differences to highlight the major contribution.